



# Air mass physio-chemical characteristics over New Delhi: Impacts on aerosol hygroscopicity and CCN formation

Zainab Arub[1], Sahil Bhandari[2], Shahzad Gani [3], Joshua S. Apte[3], Lea Hildebrandt Ruiz[2], Gazala Habib[1]

[1]Department of Civil Engineering, Indian Institute of Technology, Delhi, New Delhi, India

[2] McKetta Department of Chemical Engineering, The University of Texas at Austin, Texas, USA

[3]Department of Civil, Architectural and Environmental Engineering, The University of Texas at Austin, Texas, USA

*Correspondence to*: Zainab Arub (jyotika.mmmec@gmail.com)

**Abstract.**

This work presents for the first time long term and time-resolved estimates of hygroscopicty parameter (κ) and CCN for Delhi, a megacity that is subjected to high local anthropogenic emissions and long-range transport of pollutants. As a part of the Delhi Aerosol Supersite (DAS) campaign, characterization of aerosol composition and

size distribution were conducted from January 2017- March 2018. Air masses originating from the Arabian Sea (AS), Bay of Bengal (BB) and South Asia (SA) exhibited distinct characteristics of time-resolved sub-micron non-refractory $PM_1$ ($NRPM_1$) species, size distributions, and CCN number concentrations. SA air mass had the highest $NRPM_1$ loading with high chloride and organics followed by BB air mass which was relatively more contaminated than AS with a higher organic fraction and nitrate. The primary sources were identified as biomass-

burning, thermal power plant emissions, industrial and vehicular emissions. The average hygroscopicty parameter (κ), calculated by the mixing rule was ~ 0.3 (varying between 0.13 and 0.77) for all the air masses (0.32±0.06 for AS, 0.31±0.06 for BB and 0.32±0.10 for SA). The diurnal variations of κ were impacted by the chemical properties and thus source activities. The total, Aitken, and Accumulation mode number concentrations were higher for SA, followed by BB and AS. The mean values of estimated CCN number concentration ($N_{CCN}$, 3669-28926 $cm^{-3}$) and

the activated fraction ($a_f$, 0.19-0.87) for supersaturations varying from 0.1-0.8% also showed the same trend (SA > BB > AS). The size turned out to be more important than chemical composition directly, and the $N_{CCN}$ was governed by either the Aitken or Accumulation modes depending upon the supersaturation (SS) and critical



diameter ($D_c$). The $a_f$ was governed mainly by the Geometric Mean Diameter (GMD), and such a high $a_f$ (0.71±0.14 for the most dominant sub-branch of SA air mass (R1) at 0.4% SS) has not been seen anywhere in the world. The high $a_f$ was a consequence of very low $D_c$ (25-130 nm for SS ranging from 0.1%-0.8%) observed for Delhi. Indirectly, the chemical properties also impacted CCN and $a_f$ by impacting the diurnal patterns of Aitken and accumulation modes, κ and $D_c$. The high hygroscopic nature of aerosols, high $N_{CCN}$ and high $a_f$ can severely impact the precipitation patterns of the Indian Monsoon in Delhi, the radiation budget and the indirect effect and need to be investigated to quantify the impacts.

## 1 Introduction

High aerosol loading can have huge climatic repercussions on precipitation including land surface feedback through rainfall, surface energy budget and variation in latent heat atmospheric influx (Tao et al., 2012). Added CCN may nucleate a larger number of smaller droplets, which then take a longer time to coalesce into raindrops (Gunn and Phillips, 1957; Squires, 1958). A greater cloud depth indicating higher rain initiation occurs in more polluted clouds. Complete suppression of warm rain might also occur and get aggravated due to additional CCN activation above the cloud base (Braga et al., 2017). While rain suppression has been observed in case of polluted urban and industrial plumes (Rosenfeld, 2000) and smoke arising from forest fires (Rosenfeld, 1999), the precipitation tendency increases due to influx of giant CCN consisting of sea salt (Rosenfeld et al. 2002) and salt playas (Yinon et al. 2002) due to acceleration of the auto-conversion rate (Rosenfeld et al. 2008). In order to understand the impact of pollution on indirect radiative forcing and precipitation in highly polluted regions, the information on CCN number concentration is essential in Global Climate Models (GCMs) and Regional Climate Models (RCMs).

As per the fifth IPCC report (Boucher et al., 2013), the two most important factors governing CCN activation and number concentration are size followed by composition. The aerosol chemical composition impacts the aerosol hygroscopicity which impacts the critical diameter and hence CCN activation. The hygroscopicity parameter (κ) has been defined as the total water uptake ability of aerosols (Petters and Kredenweis, 2007). Further, the increase in relative humidity (RH) due to water uptake by aerosol can impact visibility (Lee et al., 2016; Liu et al., 2012), secondary particle formation (Ervens et al., 2011) and measurements of remote sensing (Wang et al., 2007; Brock et al., 2016), aerosol loading and its chemical composition (Chen et al., 2018). Hence, it is essential to determine the hygroscopicity of aerosols especially in the polluted regions of the world where these impacts are expected to be highly significant.

Although the recent precipitation data during 1950-2011 averaged over July and August for Delhi reveals a significant decreasing trend, there has been an increasing trend in the frequency of heavy rainfall events and a



decrease in the frequency of wet and rainy days when it rains for a shorter time period (Guhathakurta et al., 2015). These occurrences are most likely signatures of aerosols impacting the cloud nucleating properties which calls for detailed CCN data examination. The high uncertainties associated with radiative forcings, both direct and indirect especially at the regional level are a result of poor representation of the aerosol distributions in GCMs. This is

critical for the Indian sub-continent where the variability in aerosol microphysical properties is very high at various spatial and temporal scales. These necessitate the measurement of long term aerosol physiochemical properties, hygroscopicity parameter, and CCN estimates. Detailed CCN and κ measurements in India have been carried out in different parts of the world (Rissler et al., 2004;  Bougiatioti et al., 2012; Engelhart et al., 2011)and in India at places like Kanpur (Bhattu et al., 2015; Bhattu et al., 2014; Ram et al., 2014), Mahabaleshwar  (Leena

et al.,. 2016), and eastern  Himalayas  (Roy et al., 2017). However, no CCN measurements or estimates have been developed so far for Delhi. There has been only one study that has estimated aerosol hygroscopicty based on $PM_{2.5}$ mass, RH and visibility data (Wang et al., 2019). In this work, for the first time for Delhi, time-resolved size distribution and chemical speciation measurements have been carried out from 15th January 2017- 31st March 2018 as a part of the DAS campaign (Gani et al., 2019). The time-resolved hygroscopicity parameter and CCN

estimates have been derived using chemical speciation data from Aerosol Chemical Speciation Monitor (ACSM), and number concentration data from Scanning Mobility Particle Sizer (SMPS) measured during January 2017 to March 2018. Data has been analyzed to investigate the hypotheses: (a) the precursors to SOA formation critically impact the chemical composition over Delhi; (b) the emission sources significantly impact CCN formation by governing the size distributions and chemical composition, and thus hygroscopicity; (c) physical properties impact

CCN more compared to chemical properties directly; however, the physical properties are in turn, shaped by the chemical properties.

## 2 Methodology and Instrumentation

### 2.1 Instrumentation

For a detailed assessment of aerosol physiochemical properties, Scanning Mobility Particle Sizer (SMPS, TSI, Shoreview, MN), Aerosol Chemical Speciation Monitor (ACSM, Aerodyne Research, Billerica MA) and Aethalometer (Magee Scientific Model AE33, Berkeley, CA) were operated at Indian Institute of Technology, Delhi in Block 5 at a height of nearly 15 m as a part of the Delhi Aerosol Supersite (DAS) campaign. This sampling site at New Delhi was free from any source activity except for a road 150 m away. The IIT campus is relatively

cleaner than the rest of the city. However, it lies in the heart of the city, and the outskirts of the campus experience fresh traffic influx. The IIT campus allows only limited access to vehicles and therefore has relatively lesser traffic compared to the city in general. A temperature controlled room was used to carry out the measurements. Two separate and thermally insulated sampling lines (3/4 inch outer diameter stainless steel tubes) with flows of 3 LPM



and 2 LPM equipped with PM$_1$ cyclone, in line with a water trap and a Nafion membrane diffusion dryer (Magee Scientific sample streamdryer, Berkeley, CA) were used for 1) SMPS and ACSM in conjunction with a flow controller 2) Aethalometer respectively. A brief description of the instruments is given below. A detailed description of the instruments can be found in Gani et al., 2018 and Bhandari et al., 2019.

The SMPS comprised of a differential mobility analyzer (DMA, TSI 3081), an electrostatic classifier (TSI, 3080), X-ray aerosol neutralizer (TSI, 3088), and a water-based condensation particle counter (CPC, TSI 3785). The ambient air was sampled in the size range 12 nm- 560 nm, with a time difference of 135 s between two scans. The sheath to aerosol flow was 4:1 and the total flow drawn by the CPC being 1 LPM. The two dominant modes (Aitken and Accumulation) are well captured within this size range**.** ACSM sampled the inlet air at a flow rate of

0.1 LPM with a time resolution of ~1 min. The calibration procedures and data processing have been discussed in Gani et al., 2018. ACSM collected time-resolved NRPM$_1$ (Non-refractory Particulate Matter) based on species that volatilize by 600° C and included NO$_3^-$, Cl$^-$, SO$_4^{2-}$, NH$_4^+$, and organics.

## 2.2 Qualitative separation of organic aerosols as BBOA, HOA, and OOA

The composition data presented here were collected in the Delhi Aerosol Supersite (DAS) study. While the team

conducted PMF analysis on the 15 months in the dataset, they could resolve BBOA as a separate factor only in Spring 2018. They attributed this inability to resolve primary organic aerosol (POA) to separate factors hydrocarbon like organic aerosol (HOA) and biomass burning organic aerosol (BBOA) to the unit mass resolution of the instrument (Bhandari et al., 2019) and references therein). In lieu of the lack of explicit BBOA and HOA separation in all seasons, Ng et al., 2010 compilation of profiles was analyzed combined with the profiles

identified in Spring in Delhi (since they separated POA to HOA and BBOA) in the DAS study. We observed that spring 2018 profiles fall within the bounds of the uncertainty of Ng et al. compilation. Thus, Ng et al. reference profiles were utilized for source attribution of each cluster. While factor profiles can differ across the world, taking regionally relevant profiles together with those usually employed as reference profiles for PMF analysis likely accounts for this variability. As a part of the analysis conducted here, we utilize both the mean strength at the

relevant m/z(s) (m/z 57 and 60) as well as the standard deviation (S.D.) of the profiles at these m/z(s) in the analysis.

Organic aerosols were qualitatively segregated by comparing the m/z ratios of f57 and f60 with the reference profiles of BBOA (f57: 0.0337±0.00884, f60: 0.025±0.00521), HOA (f57: 0.0838±0.00378, f60: 0.00227±0.00214) and OOA (f57: 0.00997±0.00786, f60: 0.00571±0.00349) as reported by Ng et al., 2010. This

was done by first calculating the cluster means of f57 and f60 for each cluster (Table 2). The HOA, BBOA and OOA residuals ( $R_{HOA}$ , $R_{BBOA}$ and $R_{OOA}$ ) were then calculated based on cluster means of f57 and f60 ($CM_{f57}$ and $CM_{f60}$) with respect to the corresponding means of reference profiles ($RM_{f57}$ and $RM_{f60}$), as given below for HOA in Eq 1:



$$R_{HOA} = \sqrt{(CM_{f57\_HOA} - RM_{f57\_HOA})^2 + (CM_{f60\_HOA} - RM_{f60\_HOA})^2} \tag{1}$$

The reference residuals for HOA, BBOA, and OOA ($R_{Ref\_HOA}$, $R_{Ref\_BBOA}$, $R_{Ref\_OOA}$) were then calculated using standard deviations of reference profiles ($SD_{f57\_Ref\_HOA}, SD_{f57\_Ref\_BBOA}, SD_{f57\_Ref\_OOA}$ ) as given below for HOA in Eq 2:

$$R_{Ref\_HOA} = \sqrt{(SD_{f57\_Ref\_HOA})^2 + (SD_{f60\_Ref\_HOA})^2} \tag{2}$$

The following checks were then carried out for all clusters:(a) If $R_{HOA} < R_{Ref\_HOA}$, then aerosols are HOA, else if $R_{BBOA} < R_{Ref\_BBOA}$, then aerosols are BBOA else if $R_{OOA} < R_{Ref\_OOA}$, then aerosols are OOA else mixed, (b) If $R_{BBOA} < R_{Ref\_BBOA}$, then aerosols are BBOA, else if $R_{HOA} < R_{Ref\_HOA}$, then aerosols are HOA else if $R_{OOA} < R_{Ref\_OOA}$, then aerosols are OOA else mixed, (c) If $R_{OOA} < R_{Ref\_OOA}$, then aerosols are OOA, else if $R_{BBOA} < R_{Ref\_BBOA}$, then aerosols are BBOA else if $R_{HOA} < R_{Ref\_HOA}$, then aerosols are HOA else mixed, (d) If $R_{OOA} < R_{Ref\_OOA}$, then aerosols are OOA, else if $R_{HOA} < R_{Ref\_HOA}$, then aerosols are HOA else if $R_{BBOA} < R_{Ref\_BBOA}$, then aerosols are BBOA else mixed, (e) If $R_{HOA} < R_{Ref\_HOA}$, then aerosols are HOA, else if $R_{OOA} < R_{Ref\_OOA}$, then aerosols are OOA else if $R_{BBOA} < R_{Ref\_BBOA}$, then aerosols are BBOA else mixed and (f) If $R_{BBOA} < R_{Ref\_BBOA}$, then aerosols are BBOA, else if $R_{OOA} < R_{Ref\_OOA}$, then aerosols are OOA else if $R_{HOA} < R_{Ref\_HOA}$, then aerosols are HOA else mixed. If all 6 conditions were evaluated as the same specific category (i.e. HOA or BBOA or OOA), then that category was considered as the aerosol type for a cluster else they were considered as mixed.

### 2.3 Estimation of κ and CCN:

The ACSM data was used to calculate κ as per the following mixing rule in Eq. 3 (Petters and Kredenweis, 2007):

$$\kappa = \sum_i \epsilon_i \kappa_i \tag{3}$$

where, $\epsilon_i$ and $\kappa_i$ represent the volume fractions and individual hygroscopicity parameters of the various components. The inorganics were represented by $(NH_4)_2SO_4$, $NH_4Cl$, and $NH_4NO_3$. The organic $\kappa_i$ was taken as 0.1 (Gunthe et al., 2010; Gunthe et al., 2011; Dusek et al., 2010; Rose et al., 2011) and for all inorganic salts, $\kappa_i$ values were taken from Petters and Kredenweis, 2007. It should be noted that we assume that the κ calculated from $NRPM_1$ data of ACSM has been taken to represent the bulk hygroscopicity parameter in the absence of size-resolved measurements and is a limitation of this work. The difference due to the assumption cannot be accounted for and should be investigated in the future. However, it is reported that for κ >0.1, CCN closures within 20% can be achieved assuming bulk composition and internal mixing (Wang et al., 2010). Temperature, relative humidity (RH), and the calculated κ were then used to calculate the critical diameter (Dc) from the multi-component κ - Kohler theory (Bhattu et al., 2015). The temperature and RH data are available from the RK Puram site (~ 3-4 km



aerial distance from the measurement site) maintained by the Central Pollution Control Board, India. CCN was then estimated by integrating the size distribution obtained from SMPS above $D_c$. The CCN estimates were obtained for supersaturations (SS) = 0.1%, 0.15%, 0.2%, 0.35%, 0.4%, 0.5%, 0.6%, 0.7%, 0.75%, 0.8%, 0.85% and 1%. However, for the sake of detailed analysis, 0.1%, 0.4%, and 0.8% have been chosen. 0.1% represents the

condition when the effect of chemical composition is expected to be the highest, 0.4% represents the condition for convective clouds and 0.8% represents a high supersaturation state when almost all aerosols would have a tendency to get activated as CCN.

**2.4 Airmass characterization**

In order to characterize the air masses, the Hybrid Single Particle Lagrangian Integrated Trajectory (HYSPLIT)

model was used (Draxler and Rolph, 2003) to determine the major pathways of aerosols reaching Delhi. The 5-day back trajectory analysis was done at the receptor site at a height of 500 m. The cluster analysis was then performed seasonally to identify the cluster mean trajectories per season. These mean trajectories were then again re-clustered to identify three main clusters based on the directions of the mean cluster trajectories as the Arabian Sea (AS) branch (16.5% of total trajectories), the Bay of Bengal (BB) branch (13% of total trajectories) and the

South-Asian (SA) branch (70.5% of total trajectories). The BB branch was further classified as B (54%) and B.reg (45%), where B represents the air masses reaching the sea, while B.reg represents the air masses aligned towards reaching the Bay of Bengal but did not hit the sea. The SA branch was partitioned into L (17.5%), R1 (54%), R2 (18%) and R3 (11%) where L represents the local trajectories originating within India, and mainly from Delhi, Punjab and, Haryana, R1 represents trajectories coming from Pakistan and Afghanistan, R2 represents trajectories

originating from Iran, and R3 is representative of all trajectories beyond including a portion of South Africa, Mediterranean Sea, and Turkey. The seasonal clusters for winter, spring, summer, and monsoon of the year 2017 and winter and spring of the year 2018 have been shown in Figure S1. The re-clustering has been shown in Figure 1. All the chemical speciation data from ACSM and size distribution data from SMPS were then categorized as per the classification discussed above and have been used in the following discussion.

**2.5 Aerosol aging estimation**

The NOx, toluene and benzene data inventory for the entire campaign was taken from CPCB for RK Puram, whenever available. The data was not available for branch B. To determine the photochemical aging of aerosols, toluene and benzene concentrations were used to calculate the life (in hours) as per Nault et al., 2018 in Eq (4):

$$t = -\frac{1}{[OH] \times (k_{toulene} - k_{benzene})} \times \left( ln\left(\frac{toulene_i(t)}{benzene_i(t)}\right) - ln\left(\frac{toulene_i(o)}{benzene_i(o)}\right) \right) \qquad (4)$$





where, $[OH]$ = 1.5 x$10^6$ molecules/cm$^3$, $k_{toluene}$ = 2.3 x $10^{-12}$exp (-190/T) and $k_{benzene}$ = 1.8 x $10^{-12}$exp (340/T) (Atkinson et al., 2006) are the rate constants for each aromatic compound, $toulene_i$ $(o)$ = 1.85, $benzene_i$ $(o)$ = 2.31.

**3 Results and Discussion**

The HYSPLIT analysis reveals that the north-west direction is the most dominant direction which is representative of SA air masses, and within it, R1 is the most dominant indicating that on an overall basis, the emissions from Pakistan and Afghanistan regions and the sources en route govern Delhi's aerosol characteristics. However, the chemical signatures are potentially different for the various clusters, which explains the variation of aerosol

properties with time. Due to the different nature of sources and pathways, aerosol properties vary resulting in different hygroscopic properties and CCN forming potential. These aspects are discussed in the following sections.

**3.1 Introduction to characteristics and sources of air masses**

Out of the three main branches, the SA branch is the most anthropogenically contaminated followed by BB and AS branches as indicated by the mean NRPM$_1$ mass concentrations: 125.2±91.6, 45.9±23.3, and 32.5±20.6 μgm$^{-3}$

$^{3}$ respectively (Figure 2a). The total NRPM$_1$ loading for the SA branch follows the sequence: L < R3 < R1< R2. Amongst the SA branches, L is associated with the lowest organic (52.8±40.6 μgm$^{-3}$) and inorganic (42.1±33.1 μgm$^{-3}$) content while R2 has the maximum organic (85.4±59.8 μgm$^{-3}$) and inorganic (57.9±47.5 μgm$^{-3}$) content. A summary of the overall characteristics has been given in Table 1. L air-masses travel mainly over the Delhi-National Capital Region (NCR) region, Punjab and Haryana while R1 refers to air-masses originating from

Pakistan and Afghanistan. The prominent sources for SA air mass include metal processing industries (Haryana and Delhi NCR), coke and petroleum refining (Punjab), thermal power plants (Pakistan, Punjab and NCR Delhi), agricultural residue burning (Punjab and Haryana), soil dust (Pakistan, Punjab) (Jaiprakash et al., 2017) and coal mines in Pakistan where non-ideal burning of (NH$_4$)$_2$SO$_4$ occurs (Chakraborty et al., 2015).

On comparing the BB branches, total NRPM$_1$ for B (41.9±20.8 μgm$^{-3}$) is slightly lesser than B.reg (50.7±25.4

μgm$^{-3}$) and can be attributed to the fact that B.reg air mass does not travel over water (originates adjacent to coastline) but nevertheless is subject to its influence while B air mass travels over water and is therefore relatively cleaner. B.reg has lesser inorganic and more organic content than B.



The relatively higher abundance of aerosols of BB over AS can be attributed to both the sources and the pathways of air masses. In terms of source, the Bay of Bengal is more anthropogenically impacted than the Arabian Sea as concluded by the ICARB campaign (Kalapureddy et al., 2009). Previous studies (Nair et al., 2008a, 2008b; Moorthy et al., 2008) reported higher aerosol number concentration ($N_{CN}$), as well as Black Carbon (BC)

concentration over the Bay of Bengal than that of Arabian Sea in all size ranges within the marine boundary layer as well as the vertical column. The BB airmass travels over the Indo-Gangetic Plains and the AS air-mass travels across Western India and the desert region of Rajasthan. Based on previous emission estimates (Habib et al., 2006) the emission fluxes from fossil fuel dominate the aerosol burden over the Indo-Gangetic plains (IGP). The aerosol over IGP is largely composed of Inorganic oxidized matter (IOM) including fly ash from coal-fired power plants

and mineral matter from open crop waste burning (Habib et al., 2006). The AS air mass travels over western India and brings pollution from both fossil fuel combustion and desert dust (Habib et al., 2006).

**3.2 PM1 chemical composition of different air masses**

The $NRPM_1$ species ($NH_4^+$, $Cl^-$, $NO_3^-$, $SO_4^{2-}$, POA, and OOA) and BC vary significantly for the different air-masses both in terms of the mass of species (Figure 2c) and the diurnal patterns (Figure 3), leading to different

aerosol chemistry and chemical reactions. A summary of the average mass of each species for all air masses is detailed in Table 1. In brief, both POA and OOA followed by $NO_3^-$, $SO_4^{2-}$ and $Cl^-$ dominated the PM composition for the SA air mass, while OOA followed by $SO_4^{2-}$ and OOA was dominant for BB and AS air masses. High chloride is a special feature of the SA air mass, which is not apparent in the other two branches.

$NH_4^+$ has been assumed to be the dominant cation based on high Aerosol Neutralization Ratio (ANR) values

(mean values ranging from 0.95-0.85) calculated as per Zhang et al., 2007. Detailed ANR values have been given in Table S1. ANR values reveal that while AS, B, and L branches were completely neutralized, B.reg, R1, R2, and R3 were only partly neutralized indicating that minor components of sulphate, chloride, and nitrate may be bound to non-volatile salts such as $NaNO_3$ or $NaCl$ or $Na_2SO_4$ or are associated with organics as organosulphates, organochlorides, or organonitrates, evidence for which has been shown in a previous DAS study (Bhandari et al.,

25   2019).

To determine the dominant salts, $NH_4^+$ ions were neutralized with $SO_4^{2-}$ ions. The speciation of salts of $NH_4^+$ and $SO_4^{2-}$ was determined by the molar ratio of $NH_4^+$ to $SO_4^{2-}$ ions ($R[SO_4^{2-}]$). $R[SO_4^{2-}] > 2$ is indicative of $(NH_4)_2SO_4$ , while $1 < R[SO_4^{2-}] > 2$ indicates a mixture of  $(NH_4)_2SO_4$ and $NH_4HSO_4$ and $R[SO_4^{2-}] < 1$ indicates a mixture of $H_2SO_4$ and $NH_4HSO_4$ (Nenes et al., 1998; Asa-Awuku et al., 2011; Padro et al., 2012). For Delhi, $R[SO_4^{2-}] > 2$



was obtained for all branches indicating that $(NH_4)_2SO_4$ is present in all branches. Furthermore, the non-sulphate $NH_4^+$ ions $[ns\text{-}NH_4^+] = [NH_4^+] - 2 \times [SO_4^{2-}]$ were then checked for coupling with $Cl^-$, $NO_3^-$ and $[NO_3^- + Cl^-]$ ions jointly with $r^2$ values as a criterion (Du et al., 2010). All $r^2$ values have been detailed in Table S1. This analysis revealed that $(NH_4)_2SO_4$ is the dominant salt for AS and B branches based on $r^2_{NH4+/SO42-}$ values of (0.78 for AS,

and 0.75 for BB 0.75. $NH_4Cl$ formation for SA was confirmed by high $r^2$ value (0.90) for $ns\text{-}NH_4^+$ coupling with $Cl^-$. A similar finding has been reported by Bhandari et al., 2019 based on the coupling of the $NH_4Cl$ factor with wind direction. Coupling of $ns\text{-}NH_4^+$ with $NO_3^-$ revealed a good correlation (0.70) for B. In all cases, an increase in $r^2$ for combined $NO_3^- + Cl^-$ as compared to individual ions indicated that both $HNO_3$ and $HCl$ are synchronously neutralized by $NH_3$. If $Cl^-$ and $NO_3^-$ are present in fine mode, they are expected to be in the form of their respective

ammonium salts (Harrison and Pio, 1983). Thus, the dominating salts are $(NH_4)_2SO_4$ for AS and B.reg, $(NH_4)_2SO_4$ and $NH_4NO_3$ for B branches and $NH_4Cl$ for SA and its sub-branches.

The organic speciation revealed that AS was associated with BBOA, BB wherein both B and B.reg were associated with all 3 mixed organics and SA was associated with BBOA wherein L, R1 and R2 were associated with BBOA and R3 was associated with both HOA and BBOA.

The NOx emissions (in $\mu gm^{-3}$) for SA air mass (96.88±127.22) are the highest followed by BB (38.30±64.79) and then AS (36.71±68.13). BB is actually representative of B.reg only as NOx data for B air mass was not available and AS values were also scarce. The SA aerosols exhibited less aging (4.38±4.49 h) compared to B.reg. (11.58± 3.45 h), but both were representative of aged aerosols. Aging was not calculated for AS due to very little data availability.

**3.3 Diurnal variation of chemical species and probable sources**

**The South-Asian air mass**: This air mass ranks highest in $NH_4^+$ ion concentrations compared to other branches. Locally (i.e. for L), the source for $NH_4^+$ may be attributed to $NH_3$ gas from the nearby agricultural fields of the Indian Agricultural Research Institute (IARI) (Sharma et al., 2014). For R1, R2, R3, sharp spikes in early morning hours seen in the diurnal patterns of $NH_4^+$ indicate its formation from ammonia as a result of industrial exhausts

of untreated ammonia. This is because its diurnal variation is very similar to the diurnal of $NH_3$ emissions of industrial origin (Wang et al., 2015). A very prominent feature of the SA that makes it distinct from the other two air-masses is the presence of high chloride (L < R1 < R2 and R3). High $Cl^-$ in the SA branch can be attributed to several factors: (a) Khewra salt mines in Pakistan might contribute to high $Cl^-$ in other branches compared to L; (b) Locally, plastic burning, refuse burning and soil dispersion; (c) Biomass Burning, which is a very prominent





feature of the SA branch as indicated by f57 and f60 values, and is also indicated by the large number of fire counts from MODIS fire-count data (Bhattu et al., 2015), dominantly in Punjab and Haryana and few places in Pakistan; (e) Coal-based thermal power plants in Delhi, Punjab, Haryana and Pakistan; (f) Small and medium scale metal processing industries in Delhi, Punjab, and Haryana where HCl is used in pickling process of hot and

cold rolling of steel sheets and acid recovery from fume generation is not practiced (Jaiprakash et al., 2017).

As far as the increase in chloride with the increasing length of trajectories is concerned, it may be due to en route industrial emissions or waste incineration; however, the exact reason is not known. There is a marked similarity in the diurnal patterns of $NH_4^+$ and $Cl^-$ ions such that L < R1< R2 < R3 is true for both $[Cl^-]$ and $[NH_4^+]$ indicating the formation of $NH_4Cl$. $NH_4Cl$ may also be emitted directly from cement plants (Cheney et al., 1983) in Punjab.

The equilibrium constant for $Cl^-$ is more sensitive to ambient temperature than $NO_3^-$ as a result of which during daytime a large amount of $NH_4Cl$ dissociates to form $NH_3$ and HCl if the temperature exceeds 10°C (Kaneyasu et al., 1998). The diurnal patterns of both $[NH_4^+]$ and $[Cl^-]$ exhibited a sharp decrease after 08:00 in the morning which is obvious since $NH_4^+$ for SA air mass is mostly associated with $Cl^-$. At the same time $[NO_3^-]$ showed an increase between 9:00-10:00 and then started decreasing, but the rate of decrease was relatively lower than $NH_4Cl$,

which is expected as $NH_4NO_3$ is relatively more stable than $NH_4Cl$. During winter the ambient temperature drops slightly below 10°C in the morning hours and then increases sharply after 08:00 to reach a maximum at 14:00, then again starts decreasing and reaches around 10°C at mid-night (Gani et al., 2019). Since during winters, the air mass comes majorly from the north-west direction of SA air mass, it is evident that the formation and dissociation of $NH_4Cl$ were governed by the ambient temperature at the receptor site.

The chloride depletion during mid-day can also be attributed to sulphate substitution mechanism when sulphate formation enhances and is also marked by the ratio $R[SO_4^{2-}]$ ratio being greater than two for L, R1, R2, and R3. This is valid especially for L where $[SO_4^{2-}]$ increases significantly. However, the diurnal patterns of $[SO_4^{2-}]$ and $[NH_4^+]$ did not resemble each other, indicating that $(NH_4)_2SO_4$ may be present in small amounts but primarily $SO_4^{2-}$ is associated elsewhere. Hence, $SO_4^{2-}$ in combined form can be expressed in two ways: (a) small amounts

of $(NH_4)_2SO_4$ (b) majorly in combination with $K^+$. Thus, the sharp jump in $[SO_4^{2-}]$ in locally originated air masses in the late morning and afternoon hours may be attributed to $SO_2$ emissions. $SO_2$ emissions in India are primarily attributed to power generation plants that make use of coal combustion as the chief source (Reddy and Venkataraman, 2002) followed by transportation. Such coal-based power plants are located in the IGP, with a high concentration in Haryana. $SO_2$ dissolves readily in water and can form sulphite ion, which in the presence of

ozone can form sulphate ion (Erickson et al., 1976). $H_2SO_4$ formed from the reaction of $SO_2$ and ozone can react





with $NH_3$ to form $NH_4HSO_4$, which combines with $NH_3$ again to form $(NH_4)_2SO_4$ (Stelson et al., 1982; Seinfeld 1986). Since the ozone spikes in the morning from 10:00-16:00, more sulphate formation is seen when ozone is maximum. The diurnal variation for ozone has been explained in Gaur et al., 2014 for Kanpur where a spike in ozone levels is seen from 10:00- 16:00. Hike in sulphate concentration has also been previously observed for

foggy period in Kanpur at 10:00 due to resumption of photochemical activity after fog dissipation (Chakraborty et al., 2015). $SO_4^{2-}$ in SA branches may also combine with $K^+$ as $K^+$ is produced in biomass burning. It has already been mentioned that biomass burning is a highly significant feature of SA branches. Evidence for the presence of $K^+$ along with $SO_4^{2-}$ in accumulation mode has been reported in Fuzzi et al., 2007. $SO_4^{2-}$ and $NO_3^-$emissions may also be associated with secondary formation for R1, R2, and R3 due to industrial emissions from metal product

manufacturing industries in Punjab and Haryana, large-scale manufacturing of porcelain insulators, switchgear in Islamabad (Jaiprakash et al., 2017) and steel rolling mills in Iran, Iraq, and Turkey and Punjab.

The $NO_3^-$ levels are very high for SA. The high $NO_3^-$ in SA can be explained due to the non-ideal burning of $NH_4NO_3$ and NOx emissions due to mining equipment in the coal mines in Pakistan leading to high $NO_3^-$ formation (Chakraborty et al., 2015). It has been mentioned in Section 3.2 that NOx emissions in the SA branch

are very high. $NH_4^+$ neutralizes $NO_3^-$ simultaneously with $Cl^-$, however, the correlation of $[NO_3^-]$ with $[ns-NH_4^+]$ is moderate for SA. Therefore, it is expected that $NO_3^-$ might be associated with $K^+$ and $Na^+$ since biomass burning results in $K^+$ and $Na^+$ emissions (Fuzzi et al., 2007). It has been shown that $K^+$ and $Na^+$ exhibit a high affinity for nitrate during neutralization reactions, thus aiding in particulate nitrate formation (Bi et al., 2011). This is in addition to other nitrate sources that have been discussed above along with $SO_4^{2-}$ sources for SA.

The BC concentrations are highest for SA followed by AS and then BB. The biomass burning in SA air mass could be a major source of BC besides power plants, cement plants, local traffic, and industries. The POA emissions for SA followed the order L < R3 < R1< R2. The spikes during early morning hours and night time of the POA diurnal profile may be attributed to lower boundary layer heights during the two periods. BC and POA are well correlated ($r^2 = 0.77$) for R3, indicating that they are coming from primary emissions. The diurnal profiles

for all branches are similar which shows a decline as the day proceeds followed by an increase as the night proceeds. OOA is present significantly in all the three branches, but is maximum for SA. Its diurnal variation resembles that of $NO_3^-$ ($r^2 = 0.78$) indicative of its semi-volatile nature. OOA and $NO_3^-$ correlations are strongest for L ($r^2 = 0.91$), followed by R2 ($r^2 = 0.85$), R3 ($r^2 = 0.81$), and R1 ($r^2 = 0.75$).





**The Bay of Bengal air mass**: [Cl⁻] was lower in this air mass compared to SA. For both of the BB branches, fossil fuel combustion is the most likely source of Cl⁻ as fossil fuel emissions dominate IGP. Cl⁻ for BB is not correlated with ns-$NH_4^+$ and may probably be present in the form of methyl chloride, methylene chloride, carbon tetrachloride and tetrachloroethene (Ho et al., 2004).

Fossil fuel combustion from coal plants along IGP can be explained as a common source for both $SO_4^{2-}$ and $NH_4^+$ ions, leading to $(NH_4)_2SO_4$ formation ($r^2_{NH4+/SO42-} = 0.78$). This is also seen in the diurnal profiles of both $[SO_4^{2-}]$ and $[NH_4^+]$ both of which exhibit a sharp spike in the early morning hours between 10:00-16:00. $SO_2$ emissions as explained for SA via photochemical oxidation by $O_3$ in combination with $NH_3$ can lead to $(NH_4)_2SO_4$ formation. For B.reg branch, the diurnal profiles of $NH_4^+$ and $SO_4^{2-}$ exhibit double spikes (M-pattern, which is a typical

feature of NOx profile for traffic emissions) during heavy traffic hours (06:00-08:00 and around 16:00-19:00), indicating $(NH_4)_2SO4$ formation. NO from automobile exhausts can also form $NH_3$ in three ways catalytic convertors (Gandhi and Shelf, 1991) which in combination with $SO_2$ formed due to pyrolysis of sulphide fuels and subsequent oxidation, can lead to $(NH_4)_2SO_4$ formation.

The correlation of $[NO_3^-]$ with [ns-$NH_4^+$] was found to be very poor for B.reg and was appreciably high for B.The

$NO_3^-$ present in B.reg might be associated with $K^+$, $Na^+$, and $Ca^{2+}$ in the accumulation mode. $NO_3^-$ has been found to be associated with the same in light-duty vehicles with oil additives as a possible source (Sodeman et al., 2005). For B, fossil fuel combustion resulting in $NO_3^-$ emissions in combination with $NH_4^+$ (Rajput et al., 2015; Pan et al., 2019) can lead to $NH_4NO_3$ formation. This is also evident from the diurnal profile of $NO_3^-$ that shows a very similar pattern to $NH_4^+$ and is expected to be in the form of $NH_4NO_3$. The diurnal profiles of $NO_3^-$ show a decline

as the temperature increases during the day and $NO_3^-$ converts back to $HNO_3$, due to its semi-volatile nature.

The BC concentration in BB air masses was considerably lower than SA. The BC in IGP can be emitted from industries (as for B), traffic (as for B.reg) and natural sources (Derwent et al., 2001). For B and B.reg, the B.reg branch was subjected to a longer duration of anthropogenic influence compared to B which also spent a considerable time on water, hence after the early morning hours, when the various fresh emissions start increasing,

the magnitude of POA for B.reg exceeds B. However, POA for BB was quite low compared to SA. For B.reg, the spike in OOA during daytime hours is very similar to that of odd oxygen ($O_3+NO_2$) for Delhi, where the $O_X$ profile for Delhi has been given in Tiwari et al., 2015, indicating its production by local photochemistry despite the increase in boundary layer height in the afternoon.



**The Arabian Sea air mass:** Chloride amounts are relatively very low for AS compared to SA. Biomass burning as indicated by f57 and f60 measurements seems to be the main $Cl^-$ contributor to AS and might be associated with $K^+$ which is also emitted along with it.

Similar to L in the case of SA and BB, the power stations in Gujarat and Rajasthan lead to $SO_2$ emissions. Since the number of power plants is relatively lesser, the concentration is much lower compared to SA and BB air masses. $SO_2$ emissions subsequently lead to $(NH_4)_2SO_4$ formation which is the main salt present in this branch and is also evident from the high correlation between the two ions. $(NH_4)_2SO_4$ may be formed both due to emissions from power plants and traffic (similar to B.reg). Traffic emissions can be understood from the M-pattern in diurnal profiles of $NH_4^+$ and $SO_4^{2-}$, though the variation is not very pronounced, and might be suppressed due to power plant emissions. The traffic signal is more clearly implied by the diurnal profile of $(NH_4)_2SO_4$ for AS as seen in Figure S2. The correlation of $[NO_3^-]$ with $[ns-NH_4^+]$ was found to be very poor for AS indicating that $NO_3^-$ might be associated with $K^+$ and $Na^+$ similar to B.reg.

Both BC and POA are quite less compared to SA. However, compared to BB, BC is slightly higher and POA is comparable. BC is likely to be of industrial origin. The POA diurnal profile is similar to the other air masses. Similar to B.reg, the OOA diurnal pattern resembles that of odd oxygen, where the odd oxygen profile has been reported in Tiwari et al., 2015.

Thus, it can be concluded that the direct emission sources and the pre-cursors ($SO_X$, $NO_X$, $NH_3$, $O_3$, and $Ox$) that lead to particulate matter formation, strongly impact the chemical properties of aerosols. The high PM1 concentration for Delhi which exceeds the National Ambient Air Quality Standards can be mitigated only by controlling both the primary emissions and precursors. In order to address the situation justly, following measures are lacking: a) Data enlisting measurements of $PM_1$ emissions from various industries in India and Asia, b) Description of the chemical constituents of aerosol that are being emitted, both qualitatively and quantitatively, c) Defining emission limits and complying with the same. The chemical properties of aerosol also impact the hygroscopicty of aerosols as will be discussed in the following section.

### 3.4 Impact of chemical composition on the hygroscopicity of air masses

This study provides the first long term estimation of measurements of aerosol hygroscopicity in the $PM_1$ regime. The mean κ was the same for all the air masses which is ~ 0.3 (0.32±0.06 for AS, 0.31±0.06 for BB, and 0.32±0.10 for SA), as has been found to be the case for the global average of 0.27± 0.21 for continental aerosols. (Andreae



and Rosenfeld, 2008; Petters and Kreidenweis, 2007; Pöschl et al., 2009; Pringle et al., 2010). However κ varies from 0.13-0.77, and there is a difference in the diurnal variation of the hygroscopicity parameter for the various air masses (Figure 4). A similar finding was observed in China with a mean κ of 0.3 and varying in the range 0.1-0.5 (Rose et al., 2010). Recently, κ of 0.42±0.07has also been reported for PM$_{2.5}$ for Delhi based on Beta

Attenuation Monitor (BAM) measurements of PM$_{2.5}$ (Wang et al., 2019), using an indirect method in the absence of direct measurements. Thus, the dependence of κ on size cannot be underestimated for Delhi and should be dealt with in the future. This points out to the fact that organic fraction is higher in the smaller size range while the inorganic fraction increases substantially with size. Similarly, an increase in mass growth factors at 97% RH has also been observed at the Slovenian coast and was measured as  6.95 for the smaller size range (0.53-1.6 μm) and

9.78 for the larger size range (1.6-5.1 μm) (Tursic et al., 2005). Aitken mode κ has been found to be 0.25 while accumulation mode κ was 0.45 for Beijing (Gunthe et al., 2011). Even in the PM$_1$ size range, while κ of 0.1 indicates secondary organic aerosol, κ has been found to vary from 0.01- 0.8 for biomass burning aerosols in lab studies (Peters et al., 2009), thus indicating that κ values for Delhi can represent both secondary formation and biomass burning. This is true for Delhi which has both POA and OOA in all the air masses, while BBOA is present

in AS and SA air masses as detailed in the preceding sections on chemical properties.

An important observation for all branches is that when the inorganic volume fraction (of dominant salt) increases (Figure S2) or during the times when κ is high (Figure 4) or when the organic volume fraction decreases (Figure S2), a dip in D$_C$ (Figure 6) is seen, implying that a larger size regime is available for activation. The diurnal variation of κ (Figure 4) follows more strongly the diurnal pattern of the dominant inorganic salts for a cluster as

is evident from Figure S2 since the hygroscopicity parameters for inorganic salts are considerably higher than that of organics. Pearson correlation coefficient (r) values between κ and the salt volume fractions reveal that the diurnal patterns of κ were governed dominantly by volume fractions of $(NH_4)_2SO_4$ (r: 0.85) for AS, moderately by $(NH_4)_2SO_4$ (r: 0.55) and $NH_4NO_3$ (r: 0.49) for BB and  dominantly by $NH_4Cl$ for SA air masses.  For the two BB branches, κ of B branch was governed dominantly by $(NH_4)_2SO_4$ (r: 0.78) and moderately by $(NH_4)_2SO_4$ (r:

0.57) and $NH_4NO_3$ (r: 0.54) for B.reg. For SA air masses, κ of R1, R2, and R3 was governed dominantly by $NH_4Cl$ (r values of 0.71, 0.89, 0.95) and jointly by $NH_4Cl$ (r: 0.65) and $NH_4NO_3$ (r: 0.73) for L.

High volume fractions of $(NH_4)_2SO_4$ and $NH_4NO_3$ may be attributed to $SO_x$, $NO_x$, and $NH_3$ emissions due to power plant emissions and traffic. In the SA sub-branches (Figure 4), the spike in κ during the early morning (07:00-08:00) exhibits the sequence: R3 > R2 > R1 >= L and the lower spike in late evening (18:00-22:00) exhibits R3

< R2 <R1 < L, and is attributed to $NH_4Cl$ formation. This implies that during morning R3 aerosols are most



hygroscopic while L aerosols are least hygroscopic, while after 09:00, L aerosols are most hygroscopic and R3 aerosols are the least hygroscopic. The flatter curve of κ can be attributed to two factors: (a) the chloride contribution of distant trajectories decreases very steeply with time compared to the local emissions, and (b) κ of L is also supplemented substantially by $NH_4NO_3$. Thus, the source activities due to which the chemical properties of aerosols are shaped impact the hygroscopicity parameter tremendously. This consequently impacts the size regime of aerosols available for activation and has been discussed in the following section.

**3.5 Impact of governing parameters on CCN estimates of air masses**

CCN number concentration ($N_{CCN}$) for SA (22526±13439) is higher compared to BB (12526±5626) and AS (11089±6650), where values (cm$^{-3}$)are given at 0.4% SS. Amongst the SA sub-branches, $N_{CCN}$ followed increasing order as L (18810±9434) < R3 (20469±10580) < R2 (23736±13739) < R1 (24053±14743) while for the B branches, the order of increase in $N_{CCN}$ (cm$^{-3}$) was B (11699± 4900) < B.reg (14088±6506), at 0.4% SS. Correspondingly, the activated fractions ($a_f$) follow the sequence: SA (0.70±0.15) < BB (0.64±0.17) < AS (0.55±0.18) wherein for SA sub-branches, R3 (0.65±0.16) < R2 (0.694±0.16 < L (0.692±0.13) < R1 (0.71±0.15) and for BB, B.reg (0.62±0.16) < B (0.65±0.18), at 0.4% SS. Mean $N_{CCN}$ and $a_f$ for all branches at 0.1%, 0.4% and 0.8% SS have been detailed in Tables 3 and 4 respectively. The total number concentrations ($N_{CN}$ in cm$^{-3}$) follow the sequence AS (20558±9654) < BB (20864±9731) < SA (31406±15168) and for SA, L (27009±11651) < R3 (30974±12223) < R1 (32772±16475) < R2 (33371±14989) and B (19025±7704) < B.reg (24333±11956). Mean $N_{CN}$ for all branches has been listed in Table S2.

High values of $N_{CCN}$ for Delhi are consistent with other polluted regions in the world. The relevant statistics for two highly polluted sites namely, Beijing and Kanpur are presented from Gunthe et al., 2011 and Bhattu et al., 2015 respectively. $N_{CCN}$ has been found to be 7660± 3460 and 900-27000 (in cm$^{-3}$) at 0.46% SS and in the range 0.18-0.6% SS respectively. The high $N_{CCN}$ is consistent with the high $N_{CN}$. Correspondingly, $N_{CN}$ has been found to be 16800±9100 and ~50000 (cm$^{-3}$). Even though high number concentrations of CCN and CN have been reported, the $a_f$ has not been found to be so high. The $a_f$ in Beijing has been found to be 0.54±0.23and 0.66±0.23 at 0.46% SS and 0.86% SS respectively. For Kanpur, $a_f$ has been reported to be ~0.018-0.54 for 0.18-0.60% SS. However for Delhi, the $a_f$ ranges from 0.19 for AS at 0.1% SS to 0.86 for R1at 0.8% SS, implying that even at low SS, considerably large number of particles are activated and at high SS, almost all particles reach the activated state. It should be noted here that the statistics for Beijing and Kanpur correspond to the range (3-900 nm) and (14.6-680 nm) while the estimates for Delhi have been given in the (10-560 nm) range. This finding is also





consistent with Wang et al., 2019 which states that for Delhi, activation of a 0.1 μm particle requires SS ~0.18± 0.015% compared to ~0.3% for Beijing, 0.28–0.31% for Asia, Africa, and South America and ~0.22% for Europe and North America. The high activated fractions of aerosol can impact the precipitation patterns in Delhi and may be responsible for the short, intense precipitation events and decrease in overall rainfall. However, no study to

date has validated this growing trend with CCN measurements or estimates and this needs to be investigated in the future.

The $a_f$ and $N_{CCN}$ for all air masses as expected increases with an increase in supersaturation. The variation of CCN and activated fraction with SS has been shown in Figures 5 and S3. The figures clearly show that even though $N_{CCN}$ for SA is far greater compared to BB and AS, the activated fractions are fairly close. B.reg has higher CCN

but close $a_f$ compared to B. Similarly, L has the lowest $N_{CCN}$ among all SA branches but highest $a_f$. These features clearly elucidate that many factors are at play and impact $N_{CCN}$ and $a_f$ differently. In order to determine the governing parameters impacting both $N_{CCN}$ and $a_f$, the diurnal patterns of $N_{CCN}$, $a_f$, $D_c$ at 0.1%, 0.4% and 0.8% SS, $N_{CN}$, $N_{Aitken}$, $N_{Accumulation}$, κ, and GMD have been shown in Figures 6, S4 and S5.

The $N_{CCN}$ curve for SA shows a sharp diurnal feature which is not as prominent for the other two. However, in

the noontime at 0.4% and 0.8% SS, while CCN dips for SA, it rises for AS and BB (Figure 6). Further, with the increase in supersaturation, the dip in CCN of SA increases such that (a) at 0.1% SS, $SA_{CCN} > AS_{CCN}$, $BB_{CCN}$, (b) at 0.4% SS, $SA_{CCN} \sim= AS_{CCN}$, $BB_{CCN}$, and (c) at 0.8% SS, $SA_{CCN} < BB_{CCN}$ and $\sim= AS_{CCN}$. The explanation for this observation is that at 0.1% SS, the CCN is being governed by the accumulation mode but at 0.4% and 0.8% SS, it is being governed more by the Aitken mode. It is the Aitken mode that dominantly governs the total number

concentration, and hence it can be said that CCN is being governed by CN at higher SS, and by the accumulation mode at low SS. This is because as supersaturation increases, $D_c$ increases. At low SS, Dc is high ($D_c > 100$ nm) almost always at 0.1% SS for all branches as shown in Figure 6d, hence the size distribution that is being integrated to get the CCN involves the accumulation mode only. At 0.4% SS, the Dc was around 40-47 nm, therefore a considerable fraction of Aitken mode and accumulation mode is available for activation while at 0.8% SS the Dc

was ~25-30 nm, therefore, the contribution of Aitken mode further increases and accumulation mode is also available as usual. These findings are also true for BB branches (B and B.reg) and SA branches (L, R1, R2, and R3) as shown in Figure S4. It is the low value of $D_C$ relative to other places that is responsible for high CCN. As explained in Section 3.4, Dc is largely associated with κ. The $D_C$ at other places such as Kanpur has been found to vary from 50-200nm for SS ranging from 0.18-0.60 (Bhattu et al., 2015), compared to which Dc for Delhi is

lower (17-142 nm for SS ranging from 0.1%-0.8%), implying a larger regime is being available for activation. It



would also be pertinent to mention here that the dip in CCN for SA during mid-day and the hike during the same for AS and BB can be attributed to the following. (a) The dip in $N_{Accumulation}$ for SA during mid-day is much more prominent compared to AS and BB. (b) During mid-day, the $N_{Aitken}$ for SA also decreases while it increases for AS and BB. Thus, the dip in $N_{CCN}$ is strengthened by the simultaneous dip of both Aitken and Accumulation

modes while the hike in $N_{CCN}$ for AS and BB is a manifestation of the dominant hike in Aitken modes. Similar features are also exhibited for sub-branches of BB and SA. A deeper insight reveals that the dip in number concentration during mid-day for SA is most aptly seen in the diurnal pattern of POA (which is the most dominant $NRPM_1$ species) and to quite a good extent in other $NRPM_1$ species barring $SO_4^{2-}$ ion. Similarly, the hike in Aitken mode for AS and BB can be attributed to $NH_4^+$, $SO_4^{2-}$ and OOA concentrations (the dominating species in the

respective branches). Thus, the source activities and trajectory pathways impact CCN concentration at the receptor site.

The diurnal pattern for $a_f$ shows a dip in the mid-day hours for all the air-masses even though $N_{CCN}$ during mid-day for AS and BB hikes during mid-day hours. The time of dip in $a_f$ (more prominent at 0.4% and 0.8% SS) occurs earlier for AS compared to BB and SA (Figure 6). This is governed by the time of dip in the Geometric

Mean Diameter (GMD) of the three branches. The GMD diurnal variation is very similar to $a_f$ and the $r^2$ values between GMD and $a_f$ as enlisted in Table S3 also point out the same. Thus, even though a dip in $D_C$ should correspond with an increase in $a_f$ and vice-versa, this does not usually happen as the change in $Dc$ is less compared to the shift in size distribution such that not only the highest number concentration values changes, the diameter at which this occurs also changes, thereby changing the number available for activation. For example, in Figure

7 two different size distributions for AS air mass at 02:00 and 11:00 have been compared. At 02:00 and 11:00, the following characteristics were noted: (a) $GMD_{02:00}$: 83.78±16.58 nm which is considerably higher than $GMD_{11:00}$: 47.18±13.01 nm, (b) $GSD_{02:00}$: 1.69±0.13 nm which is nearly the same as $GSD_{11:00}$: 1.62±0.14 nm, (c) $D_{C02:00}$: 44.45±3.83 nm which is slightly higher than $D_{C11:00}$: 43.48±2.28 nm (at 0.4% SS), (d) $CN_{02:00}$: 15849±9269 $cm^{-3}$ which is lower than $CN_{11:00}$: 25873±9840 $cm^{-3}$ (e) $CN\_Aitken_{02:00}$: 9819±5945 $cm^{-3}$ which is lower than

$CN\_Aitken_{11:00}$: 22376±9693 $cm^{-3}$, (f) $CN\_Accu_{02:00}$: 5729±3684 $cm^{-3}$ which is higher than $CN\_Accu_{11:00}$: 3311±1569 $cm^{-3}$ (g) $a_{f02:00}$: 0.75±0.12 which is considerably higher than $a_{f11:00}$:0.38± 0.17 at 0.4% SS (h) $CCN_{02:00}$: 11595±6710 $cm^{-3}$ which is higher than $CCN_{11:00}$: 8946±3899 $cm^{-3}$. The $a_f$ at 02:00 is higher than that at 11:00. $D_c$ decrease should correspond to CCN increase, but the magnitude of $\Delta D_C$ = 0.97 nm is very small. The decrease in GMD ($\Delta$GMD = 36.6 nm), on the other hand, is very high with negligible changes in GSD. At 02:00, since $D_C$ is

considerably lesser than GSD, most of the particles are counted for activation. At 11:00, since $D_C$ and GMD are



very close, nearly 50% of particles are not available for activation. The very high Aitken mode concentration at 11:00 (higher than that at 02:00) is not available for activation at both the times. Thus, $N_{CCN}$ over here is being governed by the accumulation mode which is higher at 02:00, thus making $N_{CCN}$ higher. In this scenario, $N_{CCN}$ and $a_f$ go hand in hand. However, there also exists a second possibility. For e.g., for the SA branch, CCN at 0.4%

SS at 08:00 is higher than that at 05:00, while the $a_f$ at 08:00 is lower than that at 05:00 (Figure 6). $N_{CCN}$ is governed at these times by Aitken mode which is higher at 08:00 while $a_f$ is governed by GMD which is lower at 08:00. Thereby, it is clearly established that CCN is governed by CN (dominantly by either, Aitken or Accumulation mode as the case may be), while the $a_f$ is governed by GMD. It is also hereby established that while the physical properties impact CCN directly compared to chemical properties, the physical properties are in turn

governed by the chemical properties. The chemical properties impact CCN and $a_f$ indirectly in two ways: (a) the diurnal patterns of $NRPM_1$ species impact the diurnal pattern of Aitken and Accumulation modes which impact CCN and (b) the impact of $NRPM_1$ species on $\kappa$, hence $D_c$ and subsequently CCN by impacting the size regime available for activation.

**4 Conclusion**

Long term measurements of $NRPM_1$ species and size distribution data were carried out at New Delhi. The air masses originated from SA (L, R1, R2, and R3), BB (B and B.reg) and AS. $\kappa$ was estimated using mixing rule and the bulk $\kappa$ was assumed for the entire size distribution. Using $\kappa$ and size distribution data, CCN estimates were obtained. The SA air mass was the most contaminated air-mass followed by BB and then AS. This resulted in higher $NRPM_1$, CN (both Aitken and Accumulation modes), $N_{CCN}$ and $a_f$ for SA > BB> AS. The most dominant

salts turned out to be $(NH_4)_2SO_4$ for AS and BB and $NH_4Cl$ for SA. A, B and L branches were completely neutralized while B.reg, R1, R2, and R3 were partially neutralized. The diurnal variations of $NRPM_1$ species were governed by source activities aerosol pre-cursors like $SO_x$, $NO_x$, $NH_3$, $O_3$, and $O_x$. The mean $\kappa$ ~0.3 was the same for all air-masses, with the diurnal variation of $\kappa$ being governed by chemical species and thus, source activities. The $\kappa$ diurnal trends impacted the $D_C$ diurnal trend which in turn affected the available regime for activation. The

$N_{CCN}$ diurnal patterns were driven by Accumulation mode at lower SS and Aitken mode with an increase in SS, depending upon $D_C$ which decreases with an increase in SS. The $D_C$ obtained for Delhi was lower than that seen at other places in IGP, for example, Kanpur. The activated fraction for Delhi was very high (0.71±0.15 at 0.4% SS for R1), with the means of activated fractions varying between 0.19-0.87 for SS varying from 0.1-0.8%, whereby its diurnal patterns were governed by GMD. A CCN measurement study with a CCN counter in the

future can help verify the estimates, and a closure ratio may be determined. However, in the absence of long-term

CCN counter measurements, the importance of these findings cannot be neglected. These results can serve as valuable inputs to GCMs to better quantify precipitation. The high $NRPM_1$ loading and activated fractions are bound to significantly impact precipitation over Delhi, aerosol radiation budget and indirect effects, and need to be investigated thoroughly in the future. These investigations may answer the short intense precipitation events
occurring over Delhi and the decrease in the overall rainfall over the past half-century.

*Author contributions*. LHR, JSP, GH, and ZA designed the study. ZA, SG, and SB out the data collection. ZA carried out data processing and analysis. ZA and GH carried out the interpretation of results. ZA wrote the manuscript and was assisted by SB, LHR, and GH in reviewing the manuscript.

*Competing interests*. The authors declare that they have no conflict of interest.

*Acknowledgments*. We are thankful to the Indian Institute of Technology Delhi (IITD) for institutional support. We are grateful to all students and staff members of the Aerosol Research Characterization laboratory at IITD for their constant support. We are thankful to Philip Croteau (Aerodyne Research) for always providing timely
technical support for the ACSM.

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

**List of figures**





**List of tables**

Table 1: Mean values of NRPM$_1$, BC, inorganic ions and organics (µgm$^{-3}$) for all clusters.

Table 2: Mean values of NRPM$_1$ species (µgm$^{-3}$) for all clusters.

Table 3: Cluster means of f57 and f60 values for all branches.

5 Table 4: Mean values of CCN number concentrations (cm$^{-3}$) at 0.1%, 0.4% and 0.8% SS for all clusters.

Table 5: Mean activated fractions at 0.1%, 0.4% and 0.8% SS for all clusters.

Table S1: Aerosol Neutralization Ratio and r$^2$ values between (a) [NH$_4^+$] and [SO$_4^{2-}$], (b) [ns-NH$_4^+$] and [Cl$^-$], (c) [ns-NH$_4^+$] and [NO$_3^-$] and (d) [ns-NH$_4^+$] and [Cl$^-$+NO$_3^-$] for all air masses.

Table S2: Mean CN concentrations (cm$^{-3}$) and CN in Aitken and Accumulation modes for all clusters.

10 Table S3: Summary of r$^2$ values between GMD and a$_f$ for all clusters at SS=0.1%, 0.4%, and 0.8%.



**Figure 1**: HYSPLIT grouping of cluster mean trajectories based on directions and distances of source regions. Cluster mean trajectories were obtained for all seasons and clubbed as per directions. AS branch originated from the Arabian Sea, BB branch with sub-branches B and B.reg originated from Bay of Bengal and SA branch with sub-branches L, R1, R2 and R3 from the north-west direction originated mainly in the South Asian landmass. The

5  map layer used is from "World Countries (Generalized)", by Esri, Garmin International, 2010. (https://www.arcgis.com/home/item.html?id=170b5e6529064b8d9275168687880359). Copyright by Esri, Garmin. All rights reserved. This map is an intellectual property of Esri, Garmin and used under license. Further details may be found on www.esri.com.

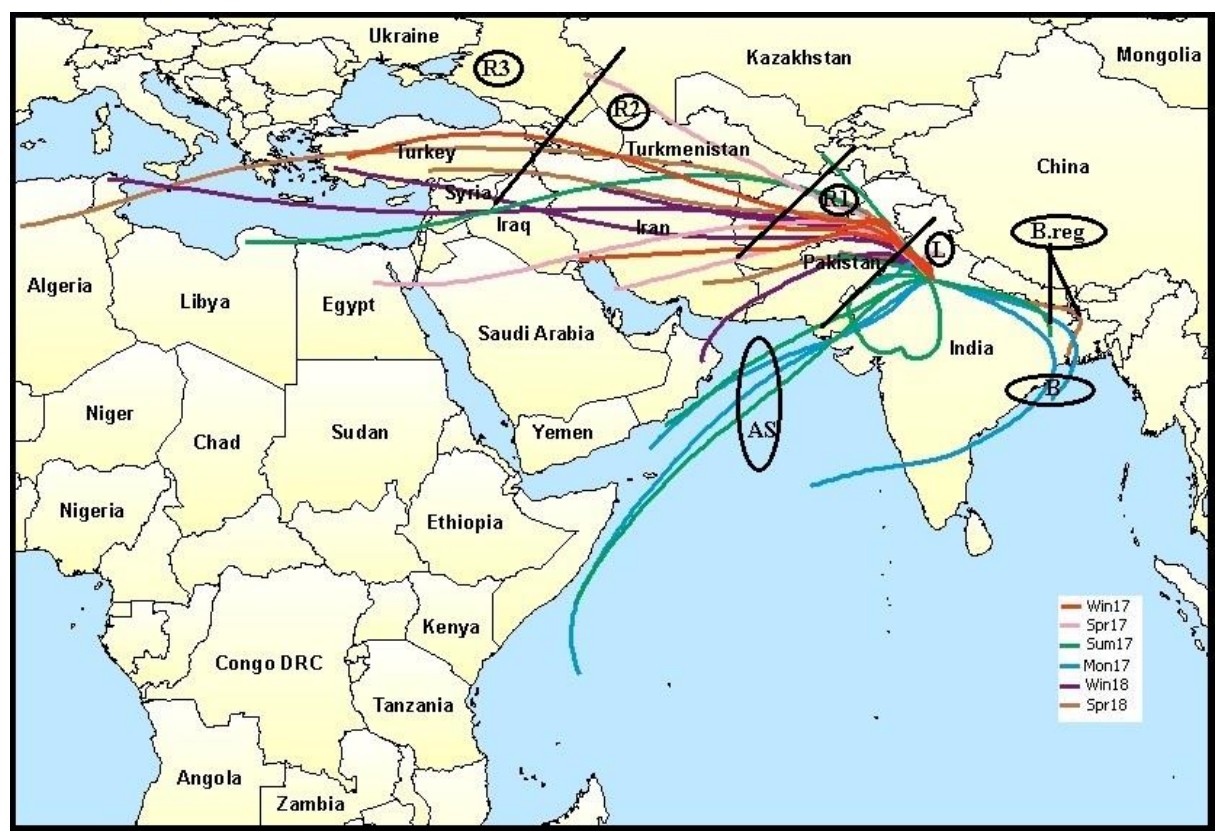





**Figure 2**: Mean values of (a) NRPM$_1$ (b) Organics, inorganics and BC (c) NRPM$_1$ species (from top to bottom) for the various air masses. The graphs on the left are for AS, BB and SA, the middle for BB branches (B and B.reg), and the right for SA branches (L, R1, R2, and R3).

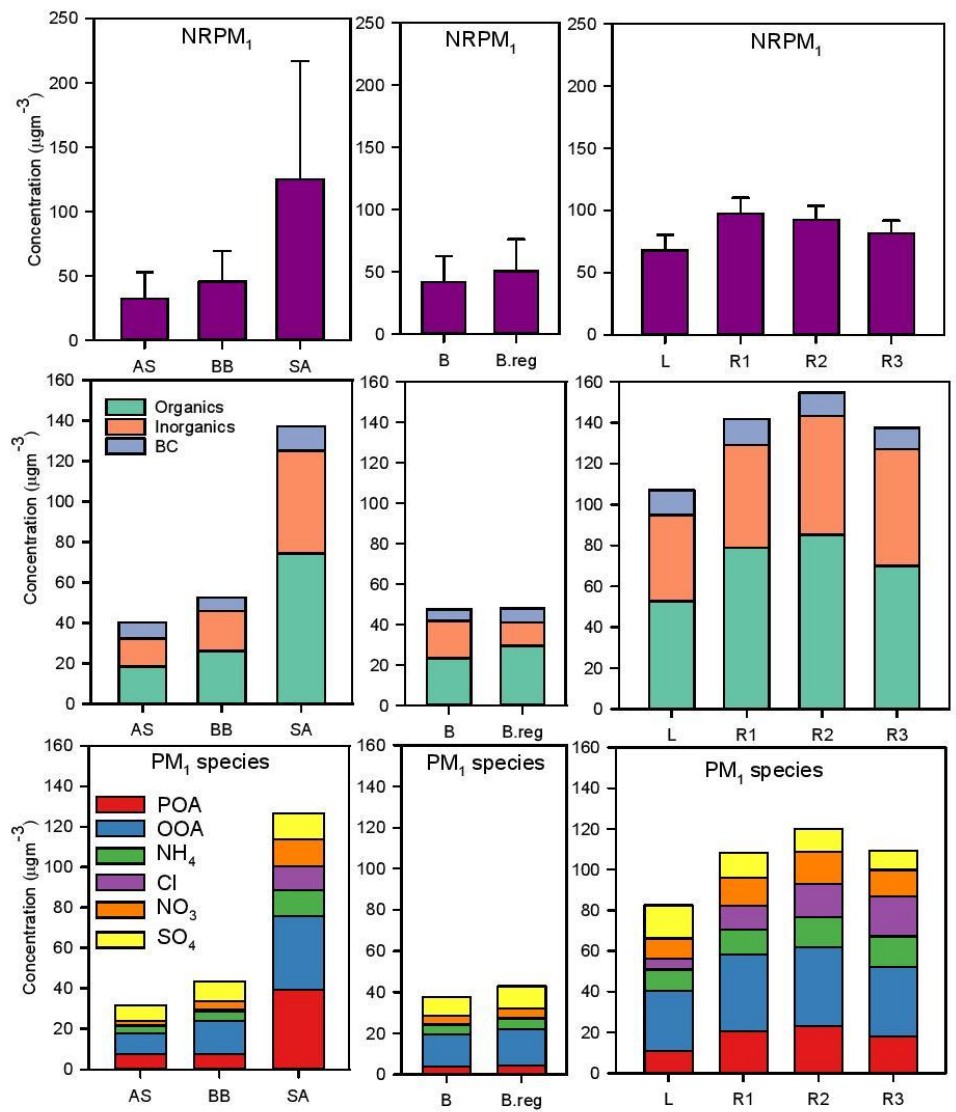





**Figure 3**: Diurnal variation of NRPM$_1$ species (NH$_4^+$, Cl$^-$, NO$_3^-$, SO$_4^{2-}$, POA, and OOA) and BC for SA (L, R1, R2, and R3) to the left, BB(B and B.reg) in the middle and AS air-masses to the right.





**Figure 4**: Diurnal variation of κ with time. The graph on the left is for AS, BB and SA, the middle for SA branches (L, R1, R2, and R3) and right is for BB branches (B and B.reg).

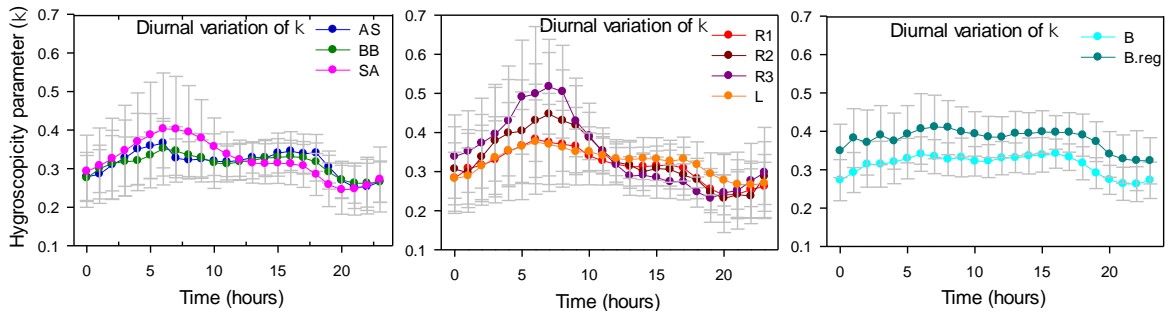



**Figure 5**: Variation of CCN and activated fraction with SS (%) for AS, BB, and SA air masses.

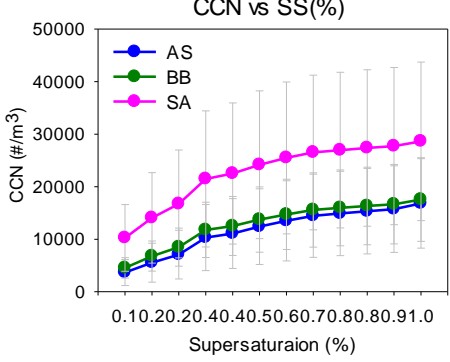
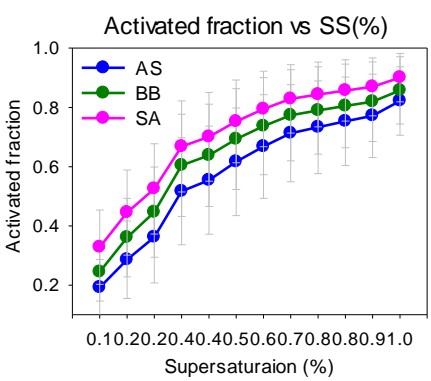



**Figure 6**: Diurnal variation of (a) $N_{CCN}$ at 0.1%, 0.4% and 0.8% SS (b) $a_f$ at 0.1%, 0.4% and 0.8% SS (c) $N_{CN}$, $N_{Aitken}$ and $N_{Accumulation}$ (d) $D_c$ at 0.1%, 0.4% and 0.8% SS (e) GMD for AS, BB and SA air masses.







**Figure7**: Comparing two size distribution profiles at different times of day for AS branch at 02:00 and 08:00.

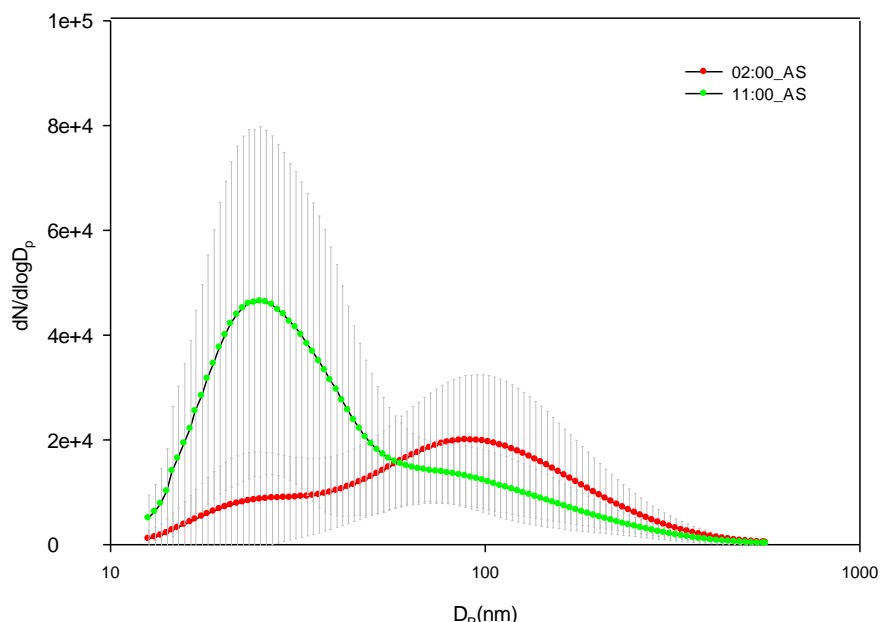





**Table 1:** Mean values of NRPM$_1$ species and BC (μgm$^{-3}$) for all clusters.

| Cluster | NRPM$_1$ | BC | NH$_4^+$ | Cl$^-$ | NO$_3^-$ | SO$_4^{2-}$ | Inorg | POA | OOA | Org |
|---------|----------|-----|----------|--------|----------|-------------|-------|-----|-----|-----|
| A | 32.5±20.6 | 7.7±7.6 | 3.6±2.5 | 0.5±1.4 | 2.3±2.9 | 7.6±4.6 | 14.0±9.3 | 7.4±8.1 | 10.3±7.0 | 18.5±13.6 |
| BB | 45.9±23.4 | 6.8±5.1 | 4.8±3.2 | 0.6±1.1 | 4.3±3.8 | 9.9±6.5 | 197±12.4 | 7.6±5.4 | 16.3±7.8 | 26.2±14.1 |
| SA | 125.2±91.6 | 12.1±10.7 | 12.7±11.1 | 12.1±20.4 | 13.3±11.8 | 12.7±9.7 | 50.8±44.0 | 39.5±43.7 | 36.3±24.2 | 74.4±58.1 |
| B | 41.9±20.7 | 5.6±5.2 | 4.6±2.6 | 0.5±0.7 | 4.2±3.8 | 9.1±5.7 | 18.5±11.0 | 7.2±4.3 | 15.5±7.3 | 23.4±11.5 |
| B.reg | 50.7±25.4 | 6.8±5.1 | 5.2±3.8 | 0.8±1.4 | 4.5±3.8 | 10.7±7.3 | 11.6±3.4 | 8.5±7.2 | 17.4±8.4 | 29.6±16.2 |
| L | 94.9±68.1 | 12.2±9.8 | 10.,4±8.2 | 5.2±12.2 | 10.1±9.7 | 16.4±9.8 | 42.0±33.0 | 18.9±23.1 | 29.5±21.7 | 52.9±40.6 |
| R1 | 129.1±97.6 | 12.6±11.3 | 12.3±10.9 | 11.6±19.4 | 13.7±12.6 | 12.5±7.6 | 50.2±44.2 | 44.2±46.7 | 37.7±26 | 79.0±6.32 |
| R2 | 143.2±92.5 | 11.4±10.3 | 14.6±12.2 | 16.4±23.6 | 15.6±11.7 | 11.3±6.7 | 57.9±47.5 | 49.4±49.5 | 38.9±22.4 | 85.4±59.9 |
| R3 | 127.1±81.6 | 10.4±8.7 | 15.0±13.4 | 19.6±26 | 13.1±10.0 | 9.3±4.8 | 56.9±49.7 | 35.7±31.8 | 34.3±18.4 | 70.2±14.7 |



**Table 2**: Cluster means of f57 and f60 values for all branches, where 'Stdev' implies standard deviation.

| Cluster | f57 | | f60 | |
| --- | --- | --- | --- | --- |
| | Mean | Stdev | Mean | Stdev |
| A | 0.02389 | 0.00777 | 0.004061 | 0.001199 |
| BB | 0.0205 | 0.006107 | 0.004294 | 0.000962 |
| SA | 0.02499 | 0.007663 | 0.007089 | 0.003569 |
| | | | | |
| B | 0.02088 | 0.005959 | 0.004368 | 0.000916 |
| B.reg | 0.01998 | 0.006266 | 0.004192 | 0.001012 |
| | | | | |
| L | 0.0239 | 0.007179 | 0.006023 | 0.002404 |
| R1 | 0.0254 | 0.007878 | 0.00717 | 0.003861 |
| R2 | 0.02509 | 0.00764 | 0.007851 | 0.003576 |
| R3 | 0.02427 | 0.006971 | 0.007084 | 0.002992 |





**Table 3**: Mean values of CCN number concentrations (cm$^{-3}$) at 0.1%, 0.4% and 0.8% SS for all clusters.

| Cluster | CCN at 0.1% SS | | CCN at 0.4% SS | | CCN at 0.8% SS | |
|---|---|---|---|---|---|---|
| | Mean | Std | Mean | Std | Mean | Std |
| A | 3669 | 2480 | 11089 | 6650 | 15339 | 8149 |
| BB | 4558 | 1945 | 12526 | 5626 | 16329 | 7385 |
| SA | 10245 | 6352 | 22526 | 13439 | 27374 | 14902 |
| B | 4469 | 1885 | 11699 | 4900 | 14892 | 5883 |
| B.reg | 4726 | 2043 | 14088 | 6506 | 19040 | 8993 |
| L | 8200 | 4612 | 18810 | 9434 | 23161 | 10845 |
| R1 | 10921 | 6843 | 24053 | 14743 | 28914 | 16265 |
| R2 | 11318 | 6071 | 23736 | 13739 | 28926 | 15111 |
| R3 | 9555 | 6077 | 20469 | 10580 | 25971 | 11963 |





**Table 4**: Mean activated fractions at 0.1%, 0.4% and 0.8% SS for all clusters.

| Cluster | $a_f$ at 0.1%SS | | $a_f$ at 0.4%SS | | $a_f$ at 0.8%SS | |
|---|---|---|---|---|---|---|
| | Mean | Std | Mean | Std | Mean | Std |
| A | 0.19 | 0.09 | 0.55 | 0.18 | 0.75 | 0.15 |
| BB | 0.25 | 0.10 | 0.64 | 0.17 | 0.81 | 0.14 |
| SA | 0.33 | 0.13 | 0.70 | 0.15 | 0.86 | 0.10 |
| B | 0.25 | 0.10 | 0.65 | 0.18 | 0.81 | 0.15 |
| B.reg | 0.23 | 0.10 | 0.62 | 0.16 | 0.80 | 0.12 |
| L | 0.31 | 0.11 | 0.69 | 0.13 | 0.85 | 0.09 |
| R1 | 0.34 | 0.13 | 0.71 | 0.15 | 0.87 | 0.10 |
| R2 | 0.35 | 0.13 | 0.69 | 0.16 | 0.85 | 0.11 |
| R3 | 0.30 | 0.13 | 0.65 | 0.16 | 0.82 | 0.12 |

