# Peer review of "Air mass physio-chemical characteristics over New Delhi: Impacts on aerosol hygroscopicity and CCN formation"

_Atmospheric Chemistry and Physics, 2019_

## Referee Comment (RC1) · Anonymous Referee #1 · 6 Jan 2020

The manuscript represents the first ever long-term measurements of non-refractory PM1 species and size distribution measurements at New Delhi, in order to gain insight of particles' hygroscopicity and the ability to act as CCN. Air masses arriving to the site were divided depending on their origin and frequency of occurrence. The respective chemical composition of the different air masses was determined, and their hygroscopicity parameter estimated. Air masses originating from South Asia were found to exhibit the highest mass loadings, featuring high chloride and organics, followed by air masses from the Bay of Bengal, featuring high organic and nitrate concentrations. For all air masses the hygroscopicity parameter was found to be around 0.3. CCN and CN concentrations were both found to follow the trend with higher concentrations from South

[Figure]

Asia, followed by the Bay of Bengal and finally the Arabian Sea. Activation fractions were also found to follow the same trend. Finally it is concluded that while the physical properties such as size impact directly CCN, they are, in turn, governed by the chemical properties.

The paper is well written and easy to follow, nevertheless there are some issues and more thorough discussion should be made in specific sections. Other than that the paper can be recommended for publication after addressing the issues listed below.

General comments:

It would be easier for the reader if all figures representing the same regions were all placed in the same order, e.g. as seen in Fig. 2 and keep the same regions in all other figures. For example, in Fig.2 SA branches (L, R1, R2, R3) are in the far right panels, while in Fig. 3 they are in far left panels and in Fig.4 they are in the middle. It may be a detail, but it would help uniformity and help the reader.

Specific comments:

- The standard deviation of $\kappa$ ($\sigma(\kappa)$) around $\kappa$ is often used as an estimate of the degree of heterogeneity (chemical dispersion) of particles (Psichoudaki et al., 2018; Lance et al., 2013). This could further be associated with the diurnal variability in the observed activation fractions as well as the chemical composition.

- Since an AE33 aethalometer was used, the contribution of BCwb can be estimated (e.g. Sandradewi et al. 2008) in order to further verify the presence of biomass burning aerosol in the SA branches (P11,L7-8 and elsewhere (e.g. P11,L20-21)). The second component (BCff) could also verify traffic emissions (e.g. P13,L8-11).

- P2,L1-3: High activation fractions as high as 0.8% at 0.38% SS have been observed at the eastern Mediterranean for air masses originating from the South (Bougiatioti et al., 2009)

- Fig.3 BB (middle panels) for BC: it seems that many points are missing in the diurnal

variability for B region, which is not commented in the text (P11,L20-28, P12,L21-28). Why is that?

- P14, L14-15: Also for the city of Athens, Greece, during wintertime when biomass burning is an important source of organic aerosol, overall hygroscopicity parameter ranged from 0.15 to 0.25 with lower values (around 0.16) being observed during night when biomass burning particles prevailed (Psichoudaki et al., 2018)

Technical corrections:

P3,L7 Detailed CCN and $\kappa$ measurements have been carried out (delete "in India")

Fig.2 (c) should read PM1 species, as now it is identical to (a)

References

Bougiatioti, A., Fountoukis, C., Kalivitis, N., Pandis, S. N., Nenes, A., and Mihalopoulos, N.: Cloud condensation nuclei measurements in the marine boundary layer of the Eastern Mediterranean: CCN closure and droplet growth kinetics, Atmos. Chem. Phys., 9, 7053-7066, https://doi.org/10.5194/acp-9-7053-2009, 2009.

Lance, S., Raatikainen, T., Onasch, T. B., Worsnop, D. R., Yu, X.-Y., Alexander, M. L., Stolzenburg, M. R., McMurry, P. H., Smith, J. N., and Nenes, A.: Aerosol mixing state, hygroscopic growth and cloud activation efficiency during MIRAGE 2006, Atmos. Chem. Phys., 13, 5049–5062, https://doi.org/10.5194/acp-13-5049-2013, 2013.

Psichoudaki, M., Nenes, A., Florou, K., Kaltsonoudis, C., & Pandis, S. N.: Hygroscopic properties of atmospheric particles emitted during wintertime biomass burning episodes in Athens. Atmospheric Environment, 178, 66–72. doi:10.1016/j.atmosenv.2018.01.004, 2018.

Sandradewi, J., Prévôt, A.S.H., Szidat, S., Perron, N., Alfarra, M.R., Lanz, V.A., Weingartner, E., and Baltensperger, U.: Using Aerosol Light Absorption Measurements for the Quantitative Determination of Wood Burning and Traffic Emission Contributions

to Particulate Matter, Environmental Science & Technology, 42 (9), 3316-3323, DOI: 10.1021/es702253m, 2008.

---

## Referee Comment (RC2) · Anonymous Referee #2 · 10 Jan 2020

Arub et al.: Air mass physio-chemical characteristics over New Delhi: Impacts on aerosol hygroscopicity and CCN formation, Atmos. Chem. Phys. Discuss., https:// doi.org/10.5194/acp-2019-1044. In review, 2019.

**Review**

**General**
The paper presents aerosol chemical composition measured with two instruments, an ACSM and an Aethalometer, and particle number size distribution measured with an SMPS from January 2017 to March 2018 during the Delhi Aerosol Supersite (DAS) campaign. The data were used for estimating the number concentrations of cloud condensation nuclei ($N_{CCN}$). The data were also classified according to geographical source areas and thereby the variabilities of aerosols in different air masses. The analysis shows also diurnal cycles of both chemical compounds and the estimated hygroscopicity parameter $\kappa$ and the estimated $N_{CCN}$. Considering the extremely high pollution levels in Delhi this type of measurements are important both for air quality and climate studies. It would be important to have measurements of $N_{CCN}$ but since no CCN counter was available it is valuable to make estimations such as in the present paper.

The paper is basically fairly easy to read and I can recommend publishing it in ACP after some additions, corrections and more detailed explanations.

At the moment there is no evaluation of the quality of the data since there was no independent $PM_1$ measurement  or CCN counter. But something can be done. A  straightforward way would be to make a closure study of mass calculated from the number size distributions measured with the SMPS and as the sum of the chemical constituents of the ACSM + BC. Doing that remember the SMPS shows size distributions using mobility diameters whereas the ACSM size range is with aerodynamic diameters. I would like to see scatter plots for the major air masses and some discussion on them.

Another thing that I noticed is that the hygroscopicity parameter was calculated using only ACSM data. There was a lot of BC in air and that definitely as also an effect on $\kappa$. Find $\kappa$(BC) from the literature and repeat the calculations taking also BC into account.

**Detailed comments**
P1,L12-13 The first sentence of the abstract  "This work presents for the first time long term and time-resolved estimates of hygroscopicity parameter ($\kappa$) and CCN for Delhi" emphasizes that the measurements were long term. That is not really true since in generally long-term measurements are such that also trends of various properties can be estimated. A bit more than one year of data cannot be considered long term. Another thing is that the  paper mainly presents variations of chemical composition, only a small part is about CCN. So it is somewhat misleading to start the abstract with the $\kappa$ and CCN.

P3,L31-32 "... relatively lesser traffic compared to the city in general ..."
"lesser" is wrong here. The comparative of little is "less", not "lesser".
        -> ... relatively less traffic than the city in general

P3, Section 2.1 This is the section showing the ACSM instrument. Write the particle size range it measures.

P4,L13 in the title 2.2, is the acronym OOA right or should it be POA?

P4,L31 What are the residuals here? Explain.

P5,L6-17 This text looks like it is taken from a computer program. Explain in a short text the contents of it and move this text to the supplement.

P5,L24-26. Present the hygroscopicity parameters of the different compounds clearly in a table and add there the references of the papers where you got them from. A compilation like that helps the readers.

P5,L32 Bhattu et al. (2015) calculated Dc using ammonium sulfate, ammonium nitrate, insoluble organics and soluble organics and gives the respective constants. You have different constituents so you should show all constants you have used either in a table or in the text like Bhattu et al. (2015) did.

P6,L1, "...CCN.." -> $N_{CCN}$

P6, Eq (4) correct toulene -> toluene

P7,L1 [OH] is definitely not constant, it varies a lot. Discuss this a bit.

P8,L19 Define aerosol neutralization ratio. Explain and give formulas.

P8,L28,  $1 < R > 2$ is never possible. Correct.

P9,L12 "...AS was associated with BBOA...". What does this mean?

P10,L15, "... ammonium nitrate is relatively more stable than ammonium chloride..." Give a reference.

P10,L20 "chloride depletion". Give formula or expain clearly what you mean here. I have calculated chloride depletion when comparing Cl-to-Na ratios in filter samples with the same ratios in pure sea salt. Now it is not possible, ACSM gives no Na concentrations.

P11,L4 "Hike in sulphate concentration...". What does hike mean here? Explain or rewrite clearly.

 P12,L14 "The correlation of NO3 with ns-NH4 was found to be very poor..." Show some scatter plots, eithier in the main text or the supplement.

P12,L21 " The BC concentration in BB air masses was considerably lower than SA." Here is a grammatical error. The sentence means that BC concentration is smaller than South Asia!
It should be " The BC concentration in BB air masses was considerably lower than in the SA air masses."
There are similar errors in several sentences in the paper. Check and correct.

P13,L5 "lesser" is a wrong word. Rewrite the sentence

P13,L13 " Both BC and POA are quite less compared to SA." I don't understand, rewrite.

P13,L17-24 This text is clearly conclusions so why don't you have it there in the "Conclusions" section?

P13, L26 A bit more than a year is not "long term".

How do the $\kappa$ values look like when you take BC into account?

P15,L19- Note that NCCN was measured with a CCN counter in the other studies and the activated fraction was actually measured. So I would be careful in making very strong conclusions.

P31,Fig 1. In the uppermost subfigures the bars should be the mean values of NRPM.
In the middle subfigs the sum of the constituents should be the same as in the upper subfigures.
But they are not. What is wrong?

---

## Author Comment (AC1) · 1 Mar 2020

**Author's response to reviewers' comments**

We are thankful to both the reviewers for their comments, suggestions and corrections that have helped improve this work. All corrections have been included in the manuscript. The reviewers' comments are written in blue and bold. The corresponding responses are written in black. The additional references have been given in the end and included in the main manuscript. Two new figures and one table have been added.

**Anonymous referee # 1**
**General comment (1)**

1. **It would be easier for the reader if all figures representing the same regions were all placed in the same order, e.g. as seen in Fig. 2 and keep the same regions in all other figures. For example, in Fig.2 SA branches (L, R1, R2, R3) are in the far right panels, while in Fig. 3 they are in far left panels and in Fig.4 they are in the middle. It may be a detail, but it would help uniformity and help the reader.**

**Response:** The order of figures has been corrected and kept the same throughout the manuscript.

**Specific comments (2-8)**

2. **The standard deviation of κ (σ(κ)) around κ is often used as an estimate of the degree of heterogeneity (chemical dispersion) of particles (Psichoudaki et al., 2018; Lance et al., 2013). This could further be associated with the diurnal variability in the observed activation fractions as well as the chemical composition.**

**Response:** We appreciate that this analysis was pointed out. The chemical dispersion for all air masses and their sub-branches have been shown in the figure below. The chemical dispersion for SA air mass during the early hours (6:00-8:00) coincided with chloride emissions and during the late night after 20:00 with POA and OOA emissions. During the time of high chloride emissions, κ also peaked since inorganics are associated with high hygroscopicity, while during the late hours, κ dropped due to an increase in organics associated with low hygroscopcity. The diurnal patterns of activated fraction, GMD and chemical dispersion were also similar. This implies that higher heterogeneity shifts GMD to a high value, thereby increasing the available regime for activation and vice-versa. There was no discernable pattern noted for the other air masses. This explanation has been included in Section 3.5. Figure 1(a) has been included in the manuscript; 1(b) and 1(c) have been included in the supplement.

**Figure1:** Diurnal variation of chemical dispersion for the various air masses (a) AS, BB and SA, (b) B and B.reg, and (c) L, R1, R2 and, R3.

[Figure]

**3. Since an AE33 aethalometer was used, the contribution of BCwb can be estimated (e.g. Sandradewi et al. 2008) in order to further verify the presence of biomass burning aerosol in the SA branches (P11,L7-8 and elsewhere (e.g. P11,L20-21)). The second component (BCff) could also verify traffic emissions (e.g. P13, L8-11).**

**Response:** Thank you for pointing out this analysis. The contribution of BCwb and BCff for all air masses was carried out (Sandradewi et al. 2008), and the following table has been included in the supplement. Since fossil fuel sources are active the year-round, there is a strong presence of BCff ranging from 70% to 86%. However, biomass burning is only active during certain specific times for short durations and is very prominent in the north-west direction for the SA air masses. It was observed that the more distant air masses exhibited a higher BCwb contribution compared to those originating within close proximity. Hence, while L was associated with 13.9% BCwb, R3 exhibited 29.2% BCwb. The BCwb contribution for A and BB air masses was 21%. It can thus be concluded that both biomass burning and traffic emissions are important sources contributing to the chemical composition of the various air masses. This analysis has been included in the manuscript.

**Table1: Contribution of BCwb and BCff for the various air masses**

| Cluster | BCwb | BCff |
|---------|------|------|
| A | 21% | 79% |
| BB | 21.60% | 78.40% |
| SA | 24.70% | 75.30% |
| | | |
| B | 26.80% | 73.20% |
| B.reg | 21.60% | 78.40% |
| | | |
| L | 13.90% | 86.10% |
| R1 | 25.20% | 74.80% |
| R2 | 29% | 71% |
| R3 | 29.20% | 70.80% |

3. **P2,L1-3: High activation fractions as high as 0.8% at 0.38% SS have been observed at the eastern Mediterranean for air masses originating from the South (Bougiatioti et al., 2009)**

**Response**: P2, L1-3 has been edited as "The $a_f$ was governed mainly by the Geometric Mean Diameter (GMD), and such a high $a_f$ (0.71±0.14 for the most dominant sub-branch of SA air mass (R1) at 0.4% SS) has not been seen anywhere in the world for a continental site." This statement was made with reference to continental locations.

4. **Fig.3 BB (middle panels) for BC: it seems that many points are missing in the diurnal variability for B region, which is not commented in the text (P11,L20-28, P12,L21-28). Why is that?**

**Response:** Unfortunately, the aethalometer was not working during that time period and hence BC data could not be collected. The unavailability of data for BC for B region has now been pointed at P12, L 22-23.

5. **P14, L14-15: Also for the city of Athens, Greece, during wintertime when biomass burning is an important source of organic aerosol, overall hygroscopicity parameter ranged from 0.15 to 0.25 with lower values (around 0.16) being observed during night when biomass burning particles prevailed (Psichoudaki et al., 2018).**

**Response**: The above detail has been added after P14, L14-15.

6. **P3,L7 Detailed CCN and _ measurements have been carried out (delete "in India") .**

**Response**: The term "in India" has been deleted.

7. **Fig.2 (c) should read PM1 species, as now it is identical to (a)**

**Response**: Fig.2(c) has been corrected to read PM1 species.

**Anonymous referee # 2**

**General comments (1-2)**

1. **At the moment there is no evaluation of the quality of the data since there was no independent PM1 measurement or CCN counter. But something can be done. A straightforward way would be to make a closure study of mass calculated from the number size distributions measured with the SMPS and as the sum of the chemical constituents of the ACSM + BC. Doing that remember the SMPS shows size distributions using mobility diameters whereas the ACSM size range is with aerodynamic diameters. I would like to see scatter plots for the major air masses and some discussion on them.**

**Response:** This analysis is indeed important. However, it has already been covered in a parallel manuscript Gani et al. 2019 that addresses the DAS (Delhi Aerosol Supersite) campaign. This detail has been added to the manuscript. The relevant scatter plot for mass closure has also been shown in the supplement (Figure S1). The following excerpt taken from Gani et al. 2019 addresses the above comment:

"*Using speciated mass concentrations and the PSD, we observed that C-PM$_1$ was highly correlated with SMPS-PM$_1$ ($R^2$=0.83), and we achieved almost complete mass closure (Fig. S1). That most of the PM$_1$ was composed of nonrefractory material and BC was consistent with past literature from Delhi which observed that metals and other nonrefractory crustal materials, which we did not measure in this study, constituted less than 5 % of PM$_1$ (Jaiprakash et al., 2017). We estimated that the C-PM$_1$ concentrations observed at our site were generally ~85 % of the PM$_{2.5}$ concentrations ($R^2$=0.54 and slope=0.85 for linear fit of hourly C-PM$_1$ and PM$_{2.5}$ concentrations over entire campaign) measured at the nearest monitoring station that is operated by the Delhi Pollution Control Committee (DPCC), R.K. Puram (3 km away), where the annual average PM$_{2.5}$ concentration for 2017 was 140 µg m$^{-3}$.*"

2. **Another thing that I noticed is that the hygroscopicity parameter was calculated using only ACSM data. There was a lot of BC in air and that definitely as also an effect on κ. Find κ (BC) from the literature and repeat the calculations taking also BC into account.**

**Response**: κ (BC) has been taken as zero in several studies in the past (Hong et al. 2014, Leng et al. 2014, Wu et al. 2013). However, even on including BC in κ calculations, the difference in κ that was introduced was on an average 10%, shifting the mean κ of 0.32 to 0.29. The BC mass fraction and volume fraction was 10% and 9% respectively. The change in κ is not significant. The impact of BC on κ has been included in Section 3.4.

**Detailed comments (3-27)**

3. **P1,L12-13 The first sentence of the abstract "This work presents for the first time long term and time-resolved estimates of hygroscopicity parameter (κ) and CCN for Delhi" emphasizes that the measurements were long term. That is not really true since in generally long-term measurements are such that also trends of various properties can be estimated. A bit more than one year of data cannot be considered long term. Another thing is that the paper mainly presents variations of chemical composition, only a small part is about CCN. So it is somewhat misleading to start the abstract with the κ and CCN.**

**Response**: The first sentence of the abstract has been modified as "Delhi is a megacity that is subjected to high local anthropogenic emissions and long-range transport of pollutants. This work presents for the first time time-resolved estimates of hygroscopicty parameter (κ) and CCN spanning for more than a year derived from chemical composition and size distribution data."

With reference to CCN and κ estimates, such studies are usually short term and span over a month. This data is hence huge as it spans for more than a year. The term "long term" has been replaced by "spanning for more than a year". In addition to it, the paper presents variations of chemical properties but it has been tried best to capture their effects

on both κ and CCN. The aim of starting the abstract with this idea is because this is the most valuable contribution of the work and is extremely relevant for the Indian sub-continent that faces a dearth of such data.

4.  **P3,L31-32 "... relatively lesser traffic compared to the city in general ..."**
    **"lesser" is wrong here. The comparative of little is "less", not "lesser".**
    **-> ... relatively less traffic than the city in general.**

**Response**: The word "lesser" has been replaced by "less".

5.  **P3, Section 2.1 This is the section showing the ACSM instrument. Write the particle size range it measures.**

**Response**: It has been mentioned in Section 2.1 that ACSM is equipped with a PM1 cyclone.

6.  **P4,L13 in the title 2.2, is the acronym OOA right or should it be POA?**

**Response**: It should be OOA as the speciation of organics has been discussed.

7.  **P4,L31 What are the residuals here? Explain.**

**Response:** Residuals represent the deviation of the cluster means from the reference profiles. The mathematical definitions have been given in Equations 1 and 2in the manuscript.

8.  **P5,L6-17 This text looks like it is taken from a computer program. Explain in a short text the contents of it and move this text to the supplement.**

**Response:** The referred text has been replaced with the following text:

 "The residuals of the cluster means were then compared with the reference residuals as per six conditions described in detail in Text S1 and classified as HOA, BBOA, OOA or mixed."

9.  **P5,L24-26. Present the hygroscopicity parameters of the different compounds clearly in a table and add there the references of the papers where you got them from. A compilation like that helps the readers.P5,L32 Bhattu et al. (2015) calculated Dc using ammonium sulfate, ammonium nitrate, insoluble organics and soluble organics and gives the respective constants. You have different constituents so you should show all constants you have used either in a table or in the text like Bhattu et al. (2015) did.**

**Response:** This has been addressed by the addition of the following text:

"$\kappa_i$ values were taken as 0.61 for $(NH_4)_2SO_4$, 1.02 for $NH_4Cl$ and 0.67 for $NH_4NO_3$ (Sullivan et al., 2009; Petters and Kredenweis 2007). The density values to estimate the volume fraction of the inorganic constituents were taken as 1770kgm$^{-3}$ for $(NH_4)_2SO_4$, 1519 kgm$^{-3}$ for $NH_4Cl$ and 1720 kgm$^{-3}$ for $NH_4NO_3$ (CRC Handbook of Physics and Chemistry, 95th Edition). The density of organics was taken as 1.5 gcm$^{-3}$ (Bougiatioti et al., 2009)."

The hygroscopicity parameter for organics has already been reported in the manuscript.

**10. P6,L1, "...CCN.." -> NCCN**

**Response:** CCN has been replaced by 'NCCN'.

**11. P6, Eq (4) correct toulene -> toluene**

**Response:** *The word 'toulene' has been corrected to 'toluene'.*

**12. P7,L1 [OH] is definitely not constant, it varies a lot. Discuss this a bit.**

**Response:** The value for OH has been taken as $1.5 \times 10^6$ molecules $cm^{-3}$ for aging calculations similar to Nault et al. 2018 for aging calculations. We agree that [OH] is not constant, however, assuming it to be constant for aging calculations is standard practice (Nault et al. 2018). As per the authors' knowledge, the detailed diurnal variation of [OH] for Delhi is not known and hence has not been commented upon. The following detail has been added to the manuscript "The [OH] concentration is not constant and varies considerably temporally and spatially, but due to unavailability of data pertaining to its variation for Delhi, it has been assumed constant for aging calculation."

**13. P8,L19 Define aerosol neutralization ratio. Explain and give formulas.**

**Response:** Aerosol Neutralization Ratio (ANR) is defined as per Zhang et al. 2007 as the normalized ratio of the measured $NH_4^+$ concentration to the $NH_4^+$ concentration needed for full neutralization of the anions. The formula for the same is given below:

$$ANR = \frac{NH_4^+ meas}{NH_4^+ neut} = \frac{(NH_4^+/18)}{(2 \times SO_4^{2-}/96) + (NO_3^-/62) + (Cl^-/35.5)}$$

This detail has been added to the manuscript.

**14. P8,L28, 1 < R > 2 is never possible. Correct.**

**Response:** This has been corrected to 1<R<2.

**15. P9,L12 "...AS was associated with BBOA...". What does this mean?**

**Response:** It implies that the organic aerosols for the AS (Arabian Sea) air mass are of biomass burning type.

**16. P10,L15, "... ammonium nitrate is relatively more stable than ammonium chloride..." Give a reference.**

**Response:** This has been explained in Kaneyasu et al., 1999.

**17. P10,L20 "chloride depletion". Give formula or explain clearly what you mean here. I have calculated chloride depletion when comparing Cl-to-Na ratios in filter samples with the same ratios in pure sea salt. Now it is not possible, ACSM gives no Na concentrations.**

**Response:** "Chloride depletion" refers to the decrease in chloride concentrations for SA air mass beginning close to mid-day. This is not with reference to Cl-to-Na ratios. To avoid ambiguity, it has been replaced with the term "reduction in chloride concentrations."

**18. P11,L4 "Hike in sulphate concentration...". What does hike mean here? Explain or rewrite clearly.**

**Response:** "Hike in sulphate concentration..." implies a peak or increase in sulphate concentration and the word 'hike' has been replaced by the word 'peak' for clarity.

**19. P12,L14 "The correlation of NO3 with ns-NH4 was found to be very poor..." Show some scatter plots, eithier in the main text or the supplement.**

**Response:** Thank you for pointing this out.The noise in the data for correlation calculation was not removed for B.reg. For both B and B.reg, ns-NH$_4^+$ correlated well with NO$_3^-$, the correlation values being 0.63 and 0.70 respectively. The scatter plots for the same have been included in the supplement and are shown below.

**Figure2**: Scatter plots between ns-NH$_4^+$ vs. NO$_3^-$ for B (left) and B.reg (right) branches.

[Figure]

This also resulted in a change in the dominant salt assumption for B.reg, which was earlier only (NH$_4$)$_2$SO$_4$. It has been changed to (NH$_4$)$_2$SO$_4$ and NH$_4$NO$_3$. Following changes have specifically been made.

1. P12, L14-L18 have been edited as under:

"The correlation of [NO$_3^-$] with [ns-NH$_4^+$] was found to be appreciably high for both B and B.reg. For both of these branches, the fossil fuel combustion resulting in NO$_3^-$ emissions in combination with NH$_4^+$ (Rajput et al., 2015; Pan et al., 2019) can lead to NH$_4$NO$_3$ formation."

2. P9, L7 has been edited as under:

"Coupling of ns-NH$_4^+$ with NO$_3^-$ revealed a good correlation for B (0.70) and B.reg (0.63)."

3. P9, L10-11 has been edited as under:

"Thus, the dominating salts are (NH$_4$)$_2$SO$_4$ for AS, (NH$_4$)$_2$SO$_4$ and NH$_4$NO$_3$ for BB air-mass and NH$_4$Cl for SA and its sub-branches."

**20. P12,L21 " The BC concentration in BB air masses was considerably lower than SA." Here is a grammatical error. The sentence means that BC concentration is smaller than South Asia! It should be " The BC concentration in BB air masses was considerably lower than in the SA air masses." There are similar errors in several sentences in the paper. Check and correct.**

**Response***:* P12, L21 has been corrected to "The BC concentration in BB air masses was considerably lower than in SA air mass". All such errors have been likewise corrected.

**21. P13,L5 "lesser" is a wrong word. Rewrite the sentence**

**Response:** The word "lesser" has been replaced by "less".

**22. P13,L13 " Both BC and POA are quite less compared to SA." I don't understand, rewrite.**

**Response:** P13, L13 implies that the BC and POA concentrations for the AS air mass are substantially low relative to their concentrations for the SA air mass. It has been replaced with the text: "Both BC and POA for AS air mass are less than in the SA air mass."

**23. P13,L17-24 This text is clearly conclusions so why don't you have it there in the "Conclusions" section?**

**Response:** P13, L17-24 has been moved to the "Conclusions" section.

**24. P13, L26 A bit more than a year is not "long term".**

**Response:** This has been discussed in response to 3.

**25. How do the κ values look like when you take BC into account?**

**Response:** This has been discussed in response to 2.

**26. P15,L19- Note that NCCN was measured with a CCN counter in the other studies and the activated fraction was actually measured. So I would be careful in making very strong conclusions.**

**Response:** The values of activated fraction for this study were found to be very high and $N_{CCN}$ values were consistent with other polluted sites in the world based on size distribution data and chemical speciation data. It is the best that could be done with the data available at hand. However, I do believe that with the actual measurements, there can be new developments and any work of science is open to criticism based on scientific evidence, and so is this work.

**27. P31,Fig 1. In the uppermost subfigures the bars should be the mean values of NRPM. In the middle subfigs the sum of the constituents should be the same as in the upper subfigures. But they are not. What is wrong?**

**Response:** Thank you for pointing this out.  The figure has been corrected and is shown below.

**Figure 3:** Mean values of (a) $NRPM_1$ (b) Organics, inorganics and BC

[Figure]

**References**

Gani, S., Bhandari, S., Seraj, S., Wang, D. S., Patel, K., Soni, P., Arub, Z., Habib, G., Hildebrandt Ruiz, L., and Apte, J. S.: Submicron aerosol composition in the world's most polluted megacity: the Delhi Aerosol Supersite study, Atmos. Chem. Phys., 19, 6843–6859, https://doi.org/10.5194/acp-19-6843-2019, 2019.

Hong, J., Häkkinen, S. A. K., Paramonov, M., Äijälä, M., Hakala, J., Nieminen, T., ... & Bilde, M.: Hygroscopicity, CCN and volatility properties of submicron atmospheric aerosol in a boreal forest environment during the summer of 2010. Atmospheric Chemistry & Physics, 14, 4733–4748, https//doi.org/10.5194/acp-14-4733-2014, 2014.

Leng, C., Zhang, Q., Tao, J., Zhang, H., Zhang, D., Xu, C., ... & Yang, X.: Impacts of new particle formation on aerosol cloud condensation nuclei (CCN) activity in Shanghai: case study. Atmos. Chem. Phys, 14(20), 11353-11365, https//doi.org/10.5194/acp-14-12499-2014, 2014.

Wu, Z. J., Poulain, L., Henning, S., Dieckmann, K., Birmili, W., Merkel, M., ... & H Herrmann, H.: Relating particle hygroscopicity and CCN activity to chemical composition during the HCCT-2010 field campaign. Atmos. Chem. Phys, 13, 7983–7996, https//doi.org/10.5194/acp-13-7983-2013, 2013.

Petters, M. D., & Kreidenweis, S. M.: A single parameter representation of hygroscopic growth and cloud condensation nucleus activity, Atmospheric Chemistry and Physics, 7(8), 1961-1971, 2007.

Bougiatioti, A., Fountoukis, C., Kalivitis, N., Pandis, S. N., Nenes, A., & Mihalopoulos, N.: Cloud condensation nuclei measurements in the marine boundary layer of the Eastern Mediterranean: CCN closure and droplet growth kinetics. Atmos. Chem. Phys., 9, 7053–7066, 2009.

Sullivan, R.C., Moore, M.J.K., Petters, M.D., Kreidenweis, S.M., Roberts, G.C., Prather, K.A.: Effect of chemical mixing state on the hygroscopicity and cloud nucleation properties of calcium mineral dust particles. Atmos. Chem. Phys., 9, 3303–3316, https://doi.org/10.5194/acp-9-3303-2009, 2009.

Haynes, W. M.: CRC handbook of chemistry and physics, 95[th] edition. CRC press, 2014.

Zhang, Q., Jimenez, J. L., Worsnop, D. R., & Canagaratna, M.: A case study of urban particle acidity and its influence on secondary organic aerosol, Environmental science & technology, 41(9), 3213-3219, https://doi.org/10.1021/es061812j, 2007.

Kaneyasu, N., Yoshikado, H., Mizuno, T., Sakamoto, K., & Soufuku, M.: Chemical forms and sources of extremely high nitrate and chloride in winter aerosol pollution in the Kanto Plain of Japan, Atmospheric Environment, 33(11), 1745-1756, https://doi.org/10.1016/S1352-2310(98)00396-3, 1999.

Nault, B. A., Campuzano-Jost, P., Day, D. A., Schroder, J. C., Anderson, B., Beyersdorf, A. J,… & Jimenez, J. L.: Secondary organic aerosol production from local emissions dominates the organic aerosol budget over Seoul, South Korea, during KORUS-AQ, Atmos. Chem. Phys., 18, 17769–17800, https://doi.org/10.5194/acp-18-17769-2018, 2018.

---

## Author Response (AR1)

**Author's response**

The reviewers' comments are written in blue and bold. The corresponding responses and changes in the manuscript are written in black.

**Anonymous referee # 1**
**General comment (1)**

1. **It would be easier for the reader if all figures representing the same regions were all placed in the same order, e.g. as seen in Fig. 2 and keep the same regions in all other figures. For example, in Fig.2 SA branches (L, R1, R2, R3) are in the far right panels, while in Fig. 3 they are in far left panels and in Fig.4 they are in the middle. It may be a detail, but it would help uniformity and help the reader.**

**Response:** The order of figures has been corrected and kept the same throughout the manuscript.

**Changes in the manuscript:** The order of figures in Figures 3 and 4, and their titles have been modified for uniformity.

**Specific comments (2-8)**

2. **The standard deviation of κ (σ(κ)) around κ is often used as an estimate of the degree of heterogeneity (chemical dispersion) of particles (Psichoudaki et al., 2018; Lance et al., 2013). This could further be associated with the diurnal variability in the observed activation fractions as well as the chemical composition.**

**Response:** We appreciate that this analysis was pointed out. Figure 1shows the diurnal variation of chemical dispersion for the various air-masses. The chemical dispersion analysis has been explained below.

"At this junction, it would also be pertinent to mention how chemical dispersion and parameters governing CCN are inter-connected. The standard deviation of κ (σ(κ)) around κ is often used as an estimate of the degree of heterogeneity (chemical dispersion) of particles (Psichoudaki et al., 2018; Lance et al., 2013). The chemical dispersion for all air masses and their sub-branches have been shown in Figures 6, S4 and S5. The chemical dispersion for SA air mass during the early hours (6:00-8:00) coincided with chloride emissions and during the late night after 20:00 with POA and OOA emissions. During the time of high chloride emissions, κ also peaked since inorganics are associated with high hygroscopicity, while during the late hours, κ dropped due to an increase in organics associated with low hygroscopcity. The diurnal patterns of activated fraction, GMD and chemical dispersion were also similar. This implies that higher heterogeneity shifts GMD to a high value, thereby increasing the available regime for activation and vice-versa. There was no discernible pattern noted for the other air masses."

**Figure1:** Diurnal variation of chemical dispersion for the various air masses (a) AS, BB and SA, (b) B and B.reg, and, (c) L, R1, R2 and, R3.

[Figure]

**Changes in the manuscript:** The above text has been added to the manuscript at P18, L26-30 and P19, L1-5. The references, Psichoudaki et al., 2018 and Lance et al., 2013 have been added to the list of references. Figure 1(a) has been included as Figure 6(e), while Figures 1(b) and 1(c) have been included in the Supplement as Figures S4 and S5 respectively. The figure captions have also been amended accordingly.

**3. Since an AE33 aethalometer was used, the contribution of BCwb can be estimated (e.g. Sandradewi et al. 2008) in order to further verify the presence of biomass burning aerosol in the SA branches (P11,L7-8 and elsewhere (e.g. P11,L20-21)). The second component (BCff) could also verify traffic emissions (e.g. P13, L8-11).**

**Response:** We are thankful for pointing out this analysis. Table 1 shows the contribution of BCwb and BC ff for the various air masses. The detailed explanation has been presented below in Text 1 and Text 2.

**Text 1:**

"In order to determine the presence of biomass burning and traffic emissions, BCwb (wood-burning component) and BCff (traffic component) for all air masses were determined based on Aethalometer data as per Sandradewi et al. 2008. The contribution of BCwb and BCff have been summarized in Table S4. Since fossil fuel sources are active the year-round, there is a strong presence of BCff ranging from 70% to 86%. However, biomass burning is only active during certain specific times for short durations and is very prominent in the north-west direction for the SA air masses. It was observed that the more distant air masses exhibited a higher BCwb contribution compared to those originating within close proximity. Hence, while L was associated with 13.9% BCwb, R3 exhibited 29.2% BCwb. The BCwb contribution for A and BB air masses was 21%. It can thus be concluded that both biomass burning and traffic emissions are important sources contributing to the chemical composition of the various air masses."

**Text 2:**

"As far as the increase in chloride with the increasing length of trajectories is concerned, the most plausible explanation is biomass burning. It has been pointed out in Section 3.2 earlier that BCwb contribution increases as the air mass trajectories become distant, a feature similar to chloride emissions."

**Table1: Contribution of BCwb and BCff for the various air masses**

| Cluster | BCwb | BCff |
|---|---|---|

| | | |
|---|---|---|
| A | 21% | 79% |
| BB | 21.60% | 78.40% |
| SA | 24.70% | 75.30% |
| | | |
| B | 26.80% | 73.20% |
| B.reg | 21.60% | 78.40% |
| | | |
| L | 13.90% | 86.10% |
| R1 | 25.20% | 74.80% |
| R2 | 29% | 71% |
| R3 | 29.20% | 70.80% |

**Changes in the manuscript:** Text 1 has been added to the manuscript at P9, L26-29 and P10, L1-6. Text 2 has been added to the manuscript at P10, L22-24. The reference Sandradewi et al. 2008 has been added to the list of references. Table 1has been included in the Supplement as table S4. The table caption has been added to the list of Tables in the manuscript.

3. **P2,L1-3: High activation fractions as high as 0.8% at 0.38% SS have been observed at the eastern Mediterranean for air masses originating from the South (Bougiatioti et al., 2009)**

**Response**: This statement was made with reference to continental locations. Accordingly, the referred text has been edited as under:

"The $a_f$ was governed mainly by the Geometric Mean Diameter (GMD), and such a high $a_f$ ($0.71\pm0.14$ for the most dominant sub-branch of SA air mass (R1) at 0.4% SS) has not been seen anywhere in the world for a continental site."

**Changes in the manuscript:** The above text has been added to the manuscript at P2, L1-3.

4. **Fig.3 BB (middle panels) for BC: it seems that many points are missing in the diurnal variability for B region, which is not commented in the text (P11,L20-28, P12,L21-28). Why is that?**

**Response:** Unfortunately, the Aethalometer was not working during that time period and hence BC data could not be collected. The unavailability of data for BC for B region has now been pointed out with the addition of the following text:

"The missing points in the diurnal variability of BC for B region are on account of unavailability of Aethalometer data."

**Changes in the manuscript:** The above text has been added at P13, L 9-10.

5. **P14, L14-15: Also for the city of Athens, Greece, during wintertime when biomass burning is an important source of organic aerosol, overall hygroscopicity parameter ranged from 0.15 to 0.25 with lower values (around 0.16) being observed during night when biomass burning particles prevailed (Psichoudaki et al., 2018).**

**Response**: The above comment has been included.

**Changes in the manuscript:** The text at P14, L26-28 now reads as under:

"κ has also been found to vary from 0.15 to 0.25 with lower values (around 0.16) being observed during night when biomass burning particles prevailed during wintertime at Athens, Greece (Psichoudaki et al., 2018). Thus, κ values for Delhi can represent both secondary formation and biomass burning."

6. **P3,L7 Detailed CCN and _ measurements have been carried out (delete "in India") .**

**Response**: The term "in India" has been deleted.

**Changes in the manuscript:** The term "in India" has been deleted from the text at P3, L9.

7. **Fig.2 (c) should read PM1 species, as now it is identical to (a)**

**Response**: Fig.2(c) has been corrected to read PM1 species.

**Changes in the manuscript:** Fig.2(c) caption has been changed to "PM1 species".

**Anonymous referee # 2**

**General comments (1-2)**

1. **At the moment there is no evaluation of the quality of the data since there was no independent PM1measurement or CCN counter. But something can be done. A straightforward way would be to make a closure study of mass calculated from the number size distributions measured with the SMPS and as the sum of the chemical constituents of the ACSM + BC. Doing that remember the SMPS shows size distributions using mobility diameters whereas the ACSM size range is with aerodynamic diameters. I would like to see scatter plots for the major air masses and some discussion on them.**

**Response:** This analysis is indeed important. However, it has already been covered in a parallel manuscript Gani et al. 2019 that addresses the DAS (Delhi Aerosol Supersite) campaign. The relevant scatter plot for mass closure has also been shown in the supplement (Figure S1). The following excerpt taken from Gani et al. 2019 addresses the above comment:

"*Using speciated mass concentrations and the PSD, we observed that C-PM$_1$ was highly correlated with SMPS-PM$_1$ ($R^2$=0.83), and we achieved almost complete mass closure (Fig. S1). That most of the PM$_1$ was composed of nonrefractory material and BC was consistent with past literature from Delhi which observed that metals and other nonrefractory crustal materials, which we did not measure in this study, constituted less than 5 % of PM$_1$ (Jaiprakash et al., 2017). We estimated that the C-PM$_1$ concentrations observed at our site were generally ~85 % of the PM$_{2.5}$ concentrations ($R^2$=0.54 and slope=0.85 for linear fit of hourly C-PM$_1$ and PM$_{2.5}$ concentrations over entire campaign) measured at the nearest monitoring station that is operated by the Delhi Pollution Control Committee (DPCC), R.K. Puram (3 km away), where the annual average PM$_{2.5}$ concentration for 2017 was 140 μg m$^{-3}$.*"

**Changes in the manuscript:** The following text has been added to P8, L13-14:

"Mass closure between SMPS size distribution data and sum of ACSM species together with BC was achieved ($R^2$= 0.83) as detailed in our parallel manuscript Gani et al. 2019."

2. **Another thing that I noticed is that the hygroscopicity parameter was calculated using only ACSM data. There was a lot of BC in air and that definitely as also an effect on κ. Find κ (BC) from the literature and repeat the calculations taking also BC into account.**

**Response**: The impact of BC on κ has been detailed below:

**Text 1:**

"κ for BC has been taken as zero in several studies in the past (Hong et al., 2014, Leng et al., 2014, Wu et al., 2013)."

**Text 2:**

"Including BC in κ calculations leads to a difference of 10% in κ on an average, shifting the mean κ of 0.32 to 0.29. The BC mass fraction and volume fraction were 10% and 9% respectively. Thus, change in κ due to introduction of BC was not significant."

**Changes in the manuscript:** Text 1 has been added at P5, L22-23. Text 2 has been added to P14, L12-15. The references, Hong et al., 2014, Leng et al., 2014, Wu et al., 2013 have been added to the list of references.

**Detailed comments (3-27)**

3. **P1,L12-13 The first sentence of the abstract "This work presents for the first time long term and time-resolved estimates of hygroscopicity parameter (κ) and CCN for Delhi" emphasizes that the measurements were long term. That is not really true since in generally long-term measurements are such that also trends of various properties can be estimated. A bit more than one year of data cannot be considered long term. Another thing is that the paper mainly presents variations of chemical composition, only a small part is about CCN. So it is somewhat misleading to start the abstract with the κ and CCN.**

**Response**: With reference to CCN and κ estimates, such studies are usually short term and span over a month. This data is hence huge as it spans for more than a year. The term "long term" has been replaced by "spanning for more than a year".    In addition to it, the paper presents variations of chemical properties but it has been tried best to capture their effects on both κ and CCN. The aim of starting the abstract with this idea is because this is the most valuable contribution of the work and is extremely relevant for the Indian sub-continent that faces a dearth of such data.

The first sentence of the abstract has been modified as:

"Delhi is a megacity that is subjected to high local anthropogenic emissions and long-range transport of pollutants. This work presents for the first time time-resolved estimates of hygroscopicty parameter (κ) and CCN spanning for more than a year derived from chemical composition and size distribution data."

**Changes in the manuscript**: The text within quotes has been added to P1, L12-14.

4. **P3,L31-32 "... relatively lesser traffic compared to the city in general ..."**

**"lesser" is wrong here. The comparative of little is "less", not "lesser".**

**-> ... relatively less traffic than the city in general.**

**Response**: The word "lesser" has been replaced by "less".

**Changes in the manuscript**: The word "lesser" has been replaced by "less" at P3, L33 and all its occurrences in the manuscript.

5. **P3, Section 2.1 This is the section showing the ACSM instrument. Write the particle size range it measures.**

**Response**: It has been mentioned in Section 2.1 that ACSM is equipped with a PM1 cyclone.

**Changes in the manuscript**: None

6. **P4, L13 in the title 2.2, is the acronym OOA right or should it be POA?**

**Response**: It should be OOA as the speciation of organics has been discussed.

**Changes in the manuscript**: None

7. **P4, L31 What are the residuals here? Explain.**

**Response: "**Residuals represent the deviation of the cluster means from the reference profiles." The mathematical definitions have been given in Equations 1 and 2in the manuscript.

**Changes in the manuscript:** The text below has been added at P4, L32-33.

"This was followed by evaluation of residuals. Residuals represent the deviation of the cluster means from the reference profiles."

8. **P5,L6-17 This text looks like it is taken from a computer program. Explain in a short text the contents of it and move this text to the supplement.**

**Response:** The referred text has been replaced with the following text:

**Text 1**:

"The residuals of the cluster means were then compared with the reference residuals as per six conditions described in detail in Text S1 and classified as HOA, BBOA, OOA or mixed."

**Changes in the manuscript:** Text 1 above has been added to P5, L9-10. The remaining text has been moved to the supplement.

9. **P5,L24-26. Present the hygroscopicity parameters of the different compounds clearly in a table and add there the references of the papers where you got them from. A compilation like that helps the readers.P5,L32 Bhattu et al. (2015) calculated Dc using ammonium sulfate, ammonium nitrate, insoluble organics and soluble organics and gives the respective constants. You have different constituents so you should show all constants you have used either in a table or in the text like Bhattu et al. (2015) did.**

**Response:** This has been addressed by the addition of the following text:

"$\kappa_i$ values were taken as 0.61 for $(NH_4)_2SO_4$, 1.02 for $NH_4Cl$ and 0.67 for $NH_4NO_3$ (Sullivan et al., 2009; Petters and Kredenweis 2007). The density values to estimate the volume fraction of the inorganic constituents were taken as 1770kg m$^{-3}$ for $(NH_4)_2SO_4$, 1519 kg m$^{-3}$ for $NH_4Cl$ and 1720 kgm$^{-3}$ for $NH_4NO_3$ (CRC Handbook of Physics and Chemistry, 95th Edition). The density of organics was taken as 1500 kg m$^{-3}$ (Bougiatioti et al., 2009)."

The hygroscopicity parameter for organics has already been reported in the manuscript.

**Changes in the manuscript:** The text within quotes above has been added to P5, L18-22. The references, Sullivan et al., 2009, Petters and Kredenweis, 2007, for CRC Handbook of Physics and Chemistry, 95th Edition and Bougiatioti et al., 2009have been added to the list of references.

**10. P6,L1, "...CCN.." -> NCCN**

**Response:** CCN has been replaced by 'NCCN'.

**Changes in the manuscript:** At P5, L31 CCN has been replaced by "$N_{CCN}$".

**11. P6, Eq (4) correct toulene -> toluene**

**Response:** *The word 'toulene' has been corrected to 'toluene'.*

**Changes in the manuscript:** The above spelling correction has been made to Eq. 4 and text at P6, L28-29.

**12. P7,L1 [OH] is definitely not constant, it varies a lot. Discuss this a bit.**

**Response:** The value for OH has been taken as $1.5\times10^6$ molecules cm$^{-3}$ for aging calculations similar to Nault et al. 2018 for aging calculations. We agree that [OH] is not constant, however, assuming it to be constant for aging calculations is standard practice (Nault et al. 2018). As per the authors' knowledge, the detailed diurnal variation of [OH] for Delhi is not known and hence has not been commented upon.

**Changes in the manuscript:** The following text has been added to the manuscript at P6, L30 and P7, L1-2.

"The [OH] concentration is not constant and varies considerably temporally and spatially, but due to unavailability of data pertaining to its variation for Delhi, it has been assumed constant for aging calculation."

**13. P8,L19 Define aerosol neutralization ratio. Explain and give formulas.**

**Response:** Aerosol Neutralization Ratio (ANR) is explained below:

"ANR is defined as the normalized ratio of the measured $NH_4^+$ concentration to the $NH_4^+$ concentration needed for full neutralization of the anions and calculated as per Eq. (5) (Zhang et al. 2007).

$$\text{ANR}= \frac{NH_4^+ meas}{NH_4^+ neut} = \frac{(NH_4^+/18)}{(2XSO_4^{2-}/96)+(NO_3^-/62)+(Cl^-/35.5)} \tag{5}"$$

**Changes in the manuscript**: The above text within quotes has been added to P8, L22-25.

**14. P8,L28, 1 < R > 2 is never possible. Correct.**

**Response:** This has been corrected to 1<R<2.

**Changes in the manuscript**: The correction "$1 < R[SO_4^{2-}] < 2$" has been done at P9, L5.

**15. P9,L12 "...AS was associated with BBOA...". What does this mean?**

**Response:** It implies that the organic aerosols for the AS (Arabian Sea) air mass are of biomass burning type.

**Changes in the manuscript**: The text at P9, L18-20 has been modified as under:

"The organic speciation revealed that AS organics were BBOA, BB organics wherein both B organics and B.reg organics were mixed and SA organics were BBOA wherein L, R1, and R2 organics were BBOA and R3 organics included both HOA and BBOA."

**16. P10,L15, "... ammonium nitrate is relatively more stable than ammonium chloride..." Give a reference.**

**Response:** This has been explained in Kaneyasu et al., 1999.

**Changes in the manuscript:** The reference "Kaneyasu et al., 1999" has been added at P11, L3.

**17. P10,L20 "chloride depletion". Give formula or explain clearly what you mean here. I have calculated chloride depletion when comparing Cl-to-Na ratios in filter samples with the same ratios in pure sea salt. Now it is not possible, ACSM gives no Na concentrations.**

**Response:** "Chloride depletion" refers to the decrease in chloride concentrations for SA air mass beginning close to mid-day. This is not with reference to Cl-to-Na ratios. To avoid ambiguity, it has been replaced with the term "reduction in chloride concentrations."

**Changes in the manuscript:** The term "reduction in chloride concentrations" has been added to P11, L8.

**18. P11,L4 "Hike in sulphate concentration...". What does hike mean here? Explain or rewrite clearly.**

**Response:** "Hike in sulphate concentration..." implies a peak or increase in sulphate concentration and the word 'hike' has been replaced by the word 'peak' for clarity.

**Changes in the manuscript:** All occurrences of 'hike' have been replaced by 'peak' in the manuscript and also at P11, L22 as suggested.

**19. P12,L14 "The correlation of NO3 with ns-NH4 was found to be very poor..." Show some scatter plots, eithier in the main text or the supplement.**

**Response:** Thank you for pointing this out. The noise in the data for correlation calculation was not removed for B.reg. For both B and B.reg, ns-$NH_4^+$ correlated well with $NO_3^-$, the correlation values being 0.63 and 0.70 respectively.

**Figure2**: Scatter plots between ns-$NH_4^+$ vs. $NO_3^-$ for B (left) and B.reg (right) branches.

[Figure]

This also resulted in a change in the dominant salt assumption for B.reg, which was earlier only $(NH_4)_2SO_4$. It has been changed to $(NH_4)_2SO_4$ and $NH_4NO_3$.

**Changes in the manuscript**: The following changes have been made.

1. P13, L3-L5 has been edited as under:

"The correlation of $[NO_3^-]$ with $[ns-NH_4^+]$ was found to be appreciably high for both B and B.reg. For both of these branches, the fossil fuel combustion resulting in $NO_3^-$ emissions in combination with $NH_4^+$ (Rajput et al., 2015; Pan et al., 2019) can lead to $NH_4NO_3$ formation."

2. P9, L13-14 has been edited as under:

"Coupling of $ns-NH_4^+$ with $NO_3^-$ revealed a good correlation for B (0.70) and B.reg (0.63) (Figure S6)."

3. P9, L16-17 has been edited as under:

"Thus, the dominating salts are $(NH_4)_2SO_4$ for AS, $(NH_4)_2SO_4$ and $NH_4NO_3$ for BB air-mass and $NH_4Cl$ for SA and its sub-branches."

4. The scatter plots in Figure 2 have been included in the supplement as Figure S6.

**20. P12,L21 " The BC concentration in BB air masses was considerably lower than SA." Here is a grammatical error. The sentence means that BC concentration is smaller than South Asia! It should be " The BC concentration in BB air masses was considerably lower than in the SA air masses." There are similar errors in several sentences in the paper. Check and correct.**

**Response**: The suggested correction has been considered and taken care of throughout the manuscript.

**Changes in the manuscript:** P13, L9 has been corrected to "The BC concentration in BB air masses was considerably lower than in the SA air mass". All such errors have been likewise corrected.

**21. P13,L5 "lesser" is a wrong word. Rewrite the sentence**

**Response:** The word "lesser" has been replaced by "less".

**Changes in the manuscript:** "Lesser" has been replaced by "less" at P13, L22 and at all other occurrences in the manuscript.

**22. P13,L13 " Both BC and POA are quite less compared to SA." I don't understand, rewrite.**

**Response:** It implies that the BC and POA concentrations for the AS air mass are substantially low relative to their concentrations for the SA air mass. It has been replaced with the text: "Both BC and POA for AS air mass are less than in the SA air mass."

**Changes in the manuscript:** The quoted text above has been added to P14, L1.

**23. P13,L17-24 This text is clearly conclusions so why don't you have it there in the "Conclusions" section?**

**Response:** The referred text partly has been moved to the "Conclusions" section and the remaining part has been retained to interlink with the next section on hygroscopicity.

**Changes in the manuscript:** The following text has been moved to P19, L19-24 under the "Conclusions" section.

"The high PM1 concentration for Delhi which exceeds the National Ambient Air Quality Standards can be mitigated only by controlling both the primary emissions and precursors. In order to address the situation justly, following measures are lacking: a) Data enlisting measurements of $PM_1$ emissions from various industries in India and Asia, b) Description of the chemical constituents of aerosol that are being emitted, both qualitatively and quantitatively, c) Defining emission limits and complying with the same."

**24. P13, L26 A bit more than a year is not "long term".**

**Response:** This has been discussed in response to 3.

**Changes in the manuscript:** Same as in response to 3.

**25. How do the κ values look like when you take BC into account?**

**Response:** This has been discussed in response to 2.

**Changes in the manuscript:** Same as in response to 2.

**26. P15,L19- Note that NCCN was measured with a CCN counter in the other studies and the activated fraction was actually measured. So I would be careful in making very strong conclusions.**

**Response:** The values of activated fraction for this study were found to be very high and $N_{CCN}$ values were consistent with other polluted sites in the world based on size distribution data and chemical speciation data. It is the best that could be done with the data available at hand. However, I do believe that with the actual measurements, there can be new developments and any work of science is open to criticism based on scientific evidence, and so is this work.

**Changes in the manuscript:** None.

**27. P31,Fig 1. In the uppermost subfigures the bars should be the mean values of NRPM. In the middle subfigs the sum of the constituents should be the same as in the upper subfigures. But they are not. What is wrong?**

**Response:** Thank you for pointing this out. The figure has been corrected and is shown below.

**Figure 3:** Mean values of (a) $NRPM_1$ (b) Organics, inorganics and BC(c) PM1 species

[Figure]

**Changes in the manuscript:** The corrections shown in Figure 3above have been included in Figure 2.

[revised manuscript text omitted]

---

## Author Response (AR2)

**Author's response to Editor's comments**

The comments of the editor are written in blue and bold. The corresponding responses and changes in the manuscript are written in black.

1. **Rather than using bold text to emphasize sub-heading, I would recommend using sub-section titles in section 3.3: 3.3.1 The South-Asian air mass, 3.3.2 The Bay of Bengal air mass, etc.**

**Response:** Sub-section titles have been numbered as suggested.
**Changes in the manuscript:** Sub-section titles have been numbered at P10, L5, P12, L16, and P13, L15 and as per the ACP's Word Template.

2. **The many places in the text, where commas could/should be added to make the text more readable: … measurements, which was a limitation.. (p 5, l 25); … Arabian Sea, as concluded…(p 8, l 2); …2006), the emission fluxes… (p 8, l 7); …branches, indicating… (p 9, l 7); …only, as NOx… (p 9, l 22); …ions, such … , indicating… (p 10, l 25); … Pakistan, leading… (p 12, l 2); …China, with a mean… (p 14; l 17); …air masses, as detailed… (p 15, l 2); …polluted sites, namely Beijing and Kampur, are … (p 16, l 8); …for Delhi, exceeding the…Standards, can be… (p 19, l 20).**

**Response:** All commas have been added.
**Changes in the manuscript:** The commas have been added at the locations pointed out and elsewhere in the manuscript to make the text more readable.

3. **The year of publication should be in parenthesis when written as part of the text. As an example (page 4, line 6): …found in Gani et al. (2018) and Bhandari et al. (2019). The same mistake is in several places of the text.**

**Response:** All in-text references have been corrected to include parenthesis.
**Changes in the manuscript:** All in-text references have been corrected to include parenthesis at the locations pointed out and throughout the manuscript.

4. **The use of tense needs to be improved throughout the paper.**
**First, a past tense would fit better in the following places: …were carried out (p 3, l 15); Data were analyzed… (page 3, l 19); …assumed (p 5, l 24); …were chosen… (p 6, l 3); …was assumed (p 7, l 1); …was slightly (p 7, l 23); … HCL was… (p 9, l 15); …was not available (p 9, l 23); …were (p 12, lines 1 and 9); …exhibited…(p 12, l 26); … were relative low (or were very low) (p 13, l 18); …was more (p 13, l 27), etc. Second, a present tense would fit better in the following places: …is given in Table 1 (p 7, l 17); …are given (p 8, l 26); …indicate (p9, l 14); …are detailed.. (p 9, l 28); …are shown (p 16, lines 25 and 30)**

**Response:** The tense usage has been corrected.

**Changes in the manuscript:** The tense has been corrected at above mentioned places and throughout the manuscript.

**5. There are a few problematic sentences that need to rewritten to make text better understandable: lines 20-25 on page 4; lines 18-20 on page 9; lines 20-23 on page 11 (what is meant by from 10-16? during 10-16 or what?) lines 10-12 on page 14 (for this sentence, I would suggest something like …was approximately the same for all the air masses (….), and in line with the global average value of…); lines 3-5 on page 15; lines 17-19 on page 15. Please reformulate these sentences.**

**Response:** The above mentioned sentences have been rewritten and modified. Yes, from 10-16 implies "during 10-16".

**Changes in the manuscript:** The above mentioned sentences have been rewritten and modified as under:

(a) P4, L17-23: "The composition data presented in this work were collected in the Delhi Aerosol Supersite (DAS) study. PMF analysis was conducted on the 15 months in the dataset. As a result, BBOA could be resolved as a separate factor only in spring 2018. This inability to resolve primary organic aerosol (POA) to separate factors, namely hydrocarbon like organic aerosol (HOA) and biomass burning organic aerosol (BBOA), was attributed to the unit mass resolution of the instrument (Bhandari et al. (2019) and references therein). Owing to the lack of explicit BBOA and HOA separation in all seasons, Ng et al., 2010 compilation of profiles was analyzed, combined with the profiles identified in spring in Delhi."

(b) P9, L3-6: "Furthermore, the non-sulphate $NH_4^+$ ions [ns-$NH_4^+$] were calculated, as per [ns-$NH_4^+$] = [$NH_4^+$] − 2 × [$SO_4^{2-}$]. The $r^2$ values were then determined for the coupling of ns-$NH_4^+$ ions with (a) $Cl^-$ions, (b)$NO_3^-$ ions, and (c) [$NO_3^-$+ $Cl^-$] ions jointly (Du et al., 2010)."

(c) P11, L19-21: "Since the ozone spiked during the daytime (10:00-16:00), more sulphate formation was seen when ozone was maximum. The diurnal variation for ozone is explained in Gaur et al. (2014) for Kanpur where a spike in ozone levels was seen during 10:00- 16:00."

(d) P13, L19-21: "Similar to the cases of the L branch in the case of SA air mass and the BB air mass, the power stations in Gujarat and Rajasthan lead to $SO_2$ emissions. Since the power plants in this region over which the AS air mass traverses are relatively less in number, the $SO_2$concentration is much lower compared to that in the Land BB air masses."

(e) P14, L9-12: ". The mean κ was approximately the same for all the air masses which is ~ 0.3 (0.32±0.06 for AS, 0.31±0.06 for BB, and 0.32±0.10 for SA), and in line with the global average value of 0.27± 0.21 for continental aerosols (Andreae and Rosenfeld, 2008; Petters and Kreidenweis, 2007; Pöschl et al., 2009; Pringle et al., 2010)."

(f) P15, L21-23: "E.g., at 97% RH, mass growth factors of 6.95 and 9.78 were reported for the size ranges, 0.53-1.6 µm and 1.6-5.1 µm respectively, at the Slovenian coast (Tursic et al., 2005)."

**6. Section 1 reads better if a new paragraph were started from line 14, p 3 (In the work, for …).**

**Response:** A new paragraph has been started at the location pointed out.

**Changes in the manuscript:** A new paragraph has been started from P3, L15.

**7. I find the last paragraph in section 3.5 a bit odd. What do authors really mean by physical and chemical properties here. Particle size and number concentration clearly belong to former, and particle chemical composition belongs to latter, but how about particle hygroscopicity, or its ability to act as CCN? Would it simply be easier to state that aerosol physical and chemical (and optical) properties, and their time evolution, are tightly linked with each other.**

**Response:** The referred text has been modified for clarity and correctness.

**Changes in the manuscript:** The last paragraph in Section 3.5 now reads as under**:**

[revised manuscript text omitted]